# DISCRETIZATION INVARIANT LEARNING ON NEURAL FIELDS

## ABSTRACT

While neural fields have emerged as powerful representations of continuous data, there is a need for neural networks that can perform inference on such data without being sensitive to how the field is sampled, a property called discretization invariance. We develop DI-Net, a framework for learning discretization invariant operators on neural fields of any type. Whereas current theoretical analyses of discretization invariant networks are restricted to the limit of infinite samples, our analysis does not require infinite samples and establishes upper bounds on the variation in DI-Net outputs given different finite discretizations. Our framework leads to a family of neural networks driven by numerical integration via quasi-Monte Carlo sampling with discretizations of low discrepancy. DI-Nets manifest desirable theoretical properties such as universal approximation of a large class of maps between $L^2$ functions, and gradients that are also discretization invariant. DI-Nets can also be seen as generalizations of many existing network families as they bridge discrete and continuous network classes, such as convolutional neural networks (CNNs) and neural operators respectively. Experimentally, DI-Nets derived from CNNs are demonstrated to classify and segment visual data represented by neural fields under various discretizations, and sometimes even generalize to new types of discretizations at test time. Code: supplementary materials (URL to be released).

## 1 INTRODUCTION

Neural fields (NFs), which encode signals as the parameters of a neural network, have many useful properties. NFs can efficiently store and stream continuous data (Sitzmann et al., 2020b; Dupont et al., 2022; Gao et al., 2021; Takikawa et al., 2022; Cho et al., 2022), represent and render detailed 3D scenes at lightning speeds (Müller et al., 2022), and integrate data from a wide range of modalities (Gao et al., 2022). NFs are thus an appealing data representation for many applications.

However, current approaches for training networks on a dataset of NFs have major limitations. The sampling-based approach converts such data to pixels or voxels as input to discrete networks (Vora et al., 2021), but it incurs interpolation errors and does not leverage the ability to evaluate the NF anywhere on its domain. The hypernetwork approach trains a model to predict NF parameters (or a lower dimensional "modulation" of such parameters) which can be tailored for downstream tasks (Tancik et al., 2020a; Dupont et al., 2022; Mehta et al., 2021), but hypernetworks based on the parameter space of one type of NF are incompatible with other types. Moreover, hypernetworks are unsuitable for important classes of NFs whose parameters extend beyond a neural network, such as those with voxel (Sun et al., 2021; Alex Yu and Sara Fridovich-Keil et al., 2021), octree (Yu et al., 2021) or hash table (Müller et al., 2022; Takikawa et al., 2022) components. We seek to strengthen the sampling-based approach with the notion of *discretization invariance*: the output of an operator that processes a continuous signal by sampling it at a set of discrete points should be largely independent of how the sample points are chosen, particularly as the number of points becomes large.

In this paper we propose the *DI-Net*, a discretization invariant neural network for learning and inference on neural fields (Fig. 1). By parameterizing layers as integrals over parametric functions of the input field, DI-Nets have access to powerful numerical integration techniques that yield strong convergence properties, including a universal approximation theorem for a wide class of maps between function spaces. DI-Nets can be applied to any type of NF, or in fact any data that can be represented as integrable functions on a bounded measurable set. Thus DI-Nets are a broad class of neural networks that encompass other continuous networks such as neural operators, and also extend

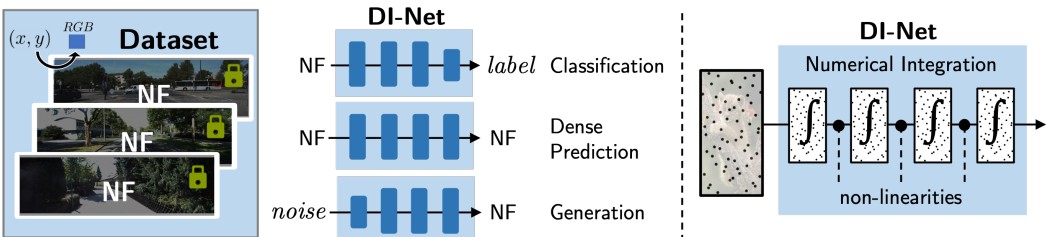

Figure 1: The DI-Net processes a neural field by evaluating it on a point set (discretization) which is used to perform numerical integration throughout the network. DI-Nets are interoperable between all types of NFs and can be trained on a broad range of tasks.

discrete networks that act on pixels, point clouds, and meshes. They can be applied to classification, segmentation, and many other tasks.

Our contributions are as follows:

- We show that discretization invariance gives rise to a family of neural networks based on numerical integration, which we call DI-Nets.
- Backpropagation through DI-Nets is discretization invariant, and they universally approximate a large class of maps between function spaces.
- DI-Nets generalize a wide class of discrete models to the continuous domain, and we derive continuous analogues of convolutional neural networks for inference on neural fields that encode visual data.
- We demonstrate convolutional DI-Nets on NF classification and dense prediction tasks, and show it can perform well under a range of discretization schemes.
- We probe the limits of discretization invariance in practice, finding that DI-Net has some ability to generalize to new discretizations at test time, modulated by the task and the type of discretizations it was trained on.

## 2   RELATED WORK

**Neural fields**   Multilayer perceptrons (MLPs) can be trained to capture a wide range of continuous data with high fidelity. The most prominent domains include shapes (Park et al., 2019; Mescheder et al., 2018), objects (Niemeyer et al., 2020; Müller et al., 2022), and 3D scenes (Mildenhall et al., 2020; Sitzmann et al., 2021), but previous works also apply NFs to gigapixel images (Martel et al., 2021), volumetric medical images (Corona-Figueroa et al., 2022), acoustic data (Sitzmann et al., 2020b; Gao et al., 2021), tactile data (Gao et al., 2022), depth and segmentation maps (Kundu et al., 2022), and 3D motion (Niemeyer et al., 2019). Hypernetworks and modulation networks were developed for learning directly with NFs, and have been demonstrated on tasks including generative modeling, data imputation, novel view synthesis and classification (Sitzmann et al., 2020b; 2021; Tancik et al., 2020a; Sitzmann et al., 2019; 2020a; Mehta et al., 2021; Chan et al., 2021; Dupont et al., 2021; 2022). Hypernetworks use meta-learning to learn to produce the MLP weights of desired output NFs, while modulation networks predict modulations that can be used to transform the parameters of an existing NF or generate a new NF. Another approach for learning NF→NF maps evaluates an input NF at grid points, produces features at the same points via a U-Net, and passes interpolated features through an MLP to produce output values at arbitrary query points (Vora et al., 2021).

**Discretization invariant networks**   Networks that are agnostic to the discretization of the data domain has been explored in several contexts. Hilbert space PCA, DeepONets and neural operators learn discretization invariant maps between function spaces (Bhattacharya et al., 2020; Lu et al., 2021; Li et al., 2020; Kovachki et al., 2021b), and are tailored to solve partial differential equations efficiently. On surface meshes, DiffusionNet (Sharp et al., 2022) uses the diffusion operator to achieve convergent behavior under mesh refinement. These previous works define discretization invariance as convergent behavior in the limit of infinite sample points, but do not characterize how different discretizations yield different behaviors in the finite case. In this work, we choose a stronger

definition that bounds the difference between any two finite discretizations, which we show implies the convergence condition. We formulate discretization invariant networks on general metric spaces, which generalizes DeepONets and neural operators, then we focus on the continuous convolution as a core layer for vision applications.

**Continuous convolutions** At the core of many discretization invariant approaches is the continuous convolution, which also provides permutation invariance, translation invariance and locality. Its applications include modeling point clouds (Wang et al., 2021; Boulch, 2019), graphs (Fey et al., 2017), fluids (Ummenhofer et al., 2019), and sequential data (Romero et al., 2021b;a), where there is typically no choice of how the data should be discretized. This work focuses on the effect of different discretizations, proposes quasi-Monte Carlo as a canonical method of generating discretizations, and can produce neural fields as output.

**Approximation capabilities of neural networks** A fundamental result in approximation theory is that the set of single-layer neural networks is dense in a large space of functionals including $L^p(\mathbb{R}^n)$ (Hornik, 1991). Subsequent works designed constructive examples using various non-linear activations (Chen et al., 1995; Chen & Chen, 1993). While this result is readily extended to multi-dimensional outputs, existing approximation results for the case of infinite dimensional outputs (e.g., $L^p(\mathbb{R}^n) \to L^p(\mathbb{R}^n)$) do not explicitly characterize the contribution of data discretization to the approximation error (Bhattacharya et al., 2020; Lanthaler et al., 2022; Kovachki et al., 2021b;a).

## 3 PRINCIPLES OF DI-NETS

In this section we formalize discretization invariance, define DI-Nets, and derive properties that enable DI-Nets to serve as a general deep learning framework on continuous data.

### 3.1 NEURAL FIELDS AS FUNCTIONS

We treat NFs as integrable maps from a domain $\Omega$ to $\mathbb{R}^c$. In particular, let $\Omega$ be a bounded measurable subset of a $d$-dimensional compact metric space. The most common case is $\Omega \subset \mathbb{R}^d$, and in the Appendix we consider the case of a $d$-dimensional manifold. Denote the space of NFs as $\mathcal{F}_c = \{f_\theta : \Omega \to \mathbb{R}^c : \int_\Omega \|f_\theta\|^2 d\mu < \infty$ and $V(f_\theta) < \infty\}$, where the variation $V(f_\theta)$ measures how much an NF fluctuates over its domain.[1] The variation of a 1D function $f \in C^1([a, b])$ is given by:

$$V(f) = \int_a^b |f'(x)|dx, \tag{1}$$

and more general definitions are given in Appendix A.1. We call $\theta$ the NF's parameters, $d$ its dimensionality, and $c$ its number of channels. For example, an occupancy network (Mescheder et al., 2018) is 3-dimensional and has 1 channel. NeRF (Mildenhall et al., 2020) is 5-dimensional (3 world coordinates and 2 view angles) and has 4 channels.

The *discretization* of an NF is a point set $X \subset \Omega$ on which the field is evaluated. We say that a map $\mathcal{H} : \mathcal{F}_c \to \mathbb{R}^n$ is discretizable if it induces a map $\hat{\mathcal{H}}^X : \mathcal{F}_c \to \mathbb{R}^n$ that depends only on the input's values at $X$.[2]

### 3.2 DISCRETIZATION INVARIANCE VIA NUMERICAL INTEGRATION

We consider two notions of discretization invariance: (1) upper bounding the deviation of the map $\hat{\mathcal{H}}^X$ from $\hat{\mathcal{H}}^Y$ for any two discretizations $X, Y$, and (2) establishing convergence of $\hat{\mathcal{H}}^{X_N}$ to $\mathcal{H}$ under particular sequences $\{X_N\}_{N \in \mathbb{N}}$ of discretizations. We use the first notion to define discretization invariant maps, then characterize sequences of discretizations under which such maps converge.

**Definition 1.** *A discretizable map $\mathcal{H} : \mathcal{F}_c \to \mathbb{R}^n$ is discretization invariant if for every discretization $X$ and neural field $f_\theta$, $\left\|\mathcal{H}[f_\theta] - \hat{\mathcal{H}}^X[f_\theta]\right\|_1$ is bounded by the product of $V(f_\theta)$ and the discrepancy*

---

[1]The parameterization of multi-layer perceptrons guarantees that NFs are square integrable and of bounded variation over a compact domain.

[2]We assume that neural field evaluation yields pointwise values, i.e. the point spread function of the underlying signal is a delta function. It is possible to accommodate non-trivial point spread functions, but this is beyond the scope of this work.

*of $X$. A map $\bar{\mathcal{H}} : \mathcal{F}_c \to \mathcal{F}_n$ is discretization invariant if $\bar{\mathcal{H}}[\cdot](x)$ is discretization invariant for all $x \in \Omega$.*

This definition establishes an upper bound on the deviation between any two discretizations by a simple application of the triangle inequality. The discrepancy of a discretization is lower for dense, evenly distributed points. For a 1D point set it is given by:

$$D(\{x_i\}_{i=1}^N) = \sup_{a \le c \le d \le b} \left| \frac{|\{x_1, \ldots, x_N\} \cap [c,d]|}{N} - \frac{d-c}{b-a} \right|. \tag{2}$$

See Appendix A.1 for general definitions of discrepancy. The product of variation and discrepancy is precisely the upper bound in the celebrated Koksma–Hlawka inequality, which bounds the difference between the integral of a function $h \in L^2(\Omega)$ and its sample mean on any point set $X \subset \Omega$:

$$\left| \frac{1}{|X|} \sum_{x' \in X} h(x') - \int_\Omega h(x)\,dx \right| \le V(h)\,D(X). \tag{3}$$

This naturally leads to a family of **discretization invariant (DI) layers** which specify a parametric map on neural fields and output its sample mean. Specifically, the action of a DI layer $\mathcal{H}_\phi$ on a neural field $f_\theta$ under discretization $X$ is:

$$\hat{\mathcal{H}}_\phi^X : f_\theta \mapsto \frac{1}{|X|} \sum_{x \in X} H_\phi[f_\theta](x). \tag{4}$$

We propose two forms of $H_\phi[f]$:

- Vector-valued DI layers ($\hat{\mathcal{H}}_\phi^X : \mathcal{F}_c \to \mathbb{R}^n$) with $H_\phi[f_\theta](x) = h_\phi(x, f_\theta(x)) \in \mathbb{R}^n$. Such layers include global pooling and learned inner products, and could be used as one of the final layers in a classification network.
- NF-valued DI layers ($\hat{\mathcal{H}}_\phi^X : \mathcal{F}_c \to \mathcal{F}_n$) with $H_\phi[f_\theta](x) : x' \mapsto h_\phi(x, x', f_\theta(x), f_\theta(x')) \in \mathbb{R}^n$ for all $x' \in \Omega$.[3] Such layers include continuous convolutions and deconvolutions, self-attention, and pooling.

In each case, $h_\phi$ must be bounded and continuous in all its variables (so that outputs remain of bounded variation), differentiable w.r.t. $\phi$ (to enable backpropagation), and Gateaux differentiable w.r.t. $f_\theta$ (which will make backpropagation discretization invariant, as we discuss in Section 3.4). We consider more general discretization invariant maps in Appendix A.2.

Importantly, the functional form of $H_\phi$ does not depend on the NF parameters $\theta$ except through $f_\theta(x)$. Thus DI layers are invariant to the parameterization of the NFs that it takes as input. This property allows these layers to be applied to a mixture of NF types, which is not possible with hypernetwork or modulation-based learning approaches.

Lastly we note that many common loss functions and regularizers generalize naturally to the continuous domain as bounded continuous maps $\mathcal{F}_c \to \mathbb{R}$ (e.g., L2 regularization) or $\mathcal{F}_c \times \mathcal{F}_c \to \mathbb{R}$ (e.g., mean squared error), so the properties of DI layers extend to losses on NFs.

### 3.3 DISCRETIZATION INVARIANT NETWORKS

A DI-Net is a directed acyclic graph of the following types of layers:

- DI layers: numerical integrators mapping an NF to a new NF or vector, as defined above
- Pointwise layers: a bounded continuous scalar function applied to each point in $\Omega$; includes nonlinear activations, batch normalization, as well as addition or concatenation of NFs
- Vector decoder: a map $\mathbb{R}^n \to \mathcal{F}_c$ specified by $n$ elements of a basis on $\mathcal{F}_c$ or interpolation of $n/c$ points (Appendix C)

---

[3] Note that the action of the layer produces a neural field with parameters $(\theta, \phi)$. In Appendix F, we discuss how to reparameterize NFs formed by a sequence of DI layers to control its parameter size.

Since pointwise layers preserve an NF's property of bounded variation, DI-Net is discretization invariant. A prototypical DI-Net for classification might consist of NF-valued DI layers separated by normalization and activation layers, and end with a vector-valued DI layer followed by softmax.

Grounding our network architecture in DI layers opens up a rich toolkit of numerical integration methods. We can generate low discrepancy discretizations using quasi-Monte Carlo (QMC), a numerical integration method with favorable convergence rates compared to standard Monte Carlo (Caflisch, 1998). The QMC discretization only requires a single pass through the network, can be deterministic or pseudorandom, and can accelerate computation when the same discretization is used for multiple network layers or all fields in a minibatch. To sample from non-uniform measures, the standard Monte Carlo method with rejection sampling can be used instead. To integrate over a fixed discretization of high discrepancy, we can use a quadrature method that replaces $1/|X|$ in the sample mean with quadrature weights. Adaptive quadrature techniques can be used to attain specific error bounds at inference time, which can be valuable in applications requiring robustness or verification.

### 3.4 CONVERGENCE UNDER EQUIDISTRIBUTED DISCRETIZATIONS

We call a sequence of discretizations $\{X_N\}_{N \in \mathbb{N}}$ whose discrepancy tends to 0 as $N \to \infty$ an *equidistributed* discretization sequence.[4] By Equation (3), DI layers converge under such sequences, i.e., $\lim_{N \to \infty} \hat{\mathcal{H}}_\phi^{X_N} \equiv \mathcal{H}_\phi$, and hence forward passes through DI-Nets are also convergent. But demonstrating convergence of DI-Net's discretized gradients is less straightforward.

Consider a scalar-valued DI layer on a single-channel NF, $\mathcal{H}_\phi : \mathcal{F}_1 \to \mathbb{R}$. The derivatives of its output w.r.t. each of its weights $\phi_k$ can be shown to converge as:

$$\lim_{N \to \infty} \frac{\partial}{\partial \phi_k} \hat{\mathcal{H}}_\phi^{X_N} \equiv \frac{\partial}{\partial \phi_k} \mathcal{H}_\phi, \tag{5}$$

under any equidistributed discretization sequence $\{X_N\}_{N \in \mathbb{N}}$. Describing the derivative of the layer's output w.r.t. the input NF is more nuanced, since pointwise derivatives are not sufficient to represent backpropagation in the continuous case. We must instead use the Gateaux derivative, which describes the linear change in a map between functions given an infinitesimal change in the input function. We prove the following in Appendix B.1:

**Proposition 1.** *For every $f_\theta \in \mathcal{F}_1$ and fixed $\tilde{x} \in \Omega$, we can design a sequence of bump functions $\{\psi_{\tilde{x}}^N\}_{N \in \mathbb{N}}$ which is 1 in a small neighborhood around $\tilde{x}$ and vanishes at each $X_N \backslash \{\tilde{x}\}$, such that:*

$$\lim_{N \to \infty} \frac{\partial}{\partial f_\theta(\tilde{x})} \hat{\mathcal{H}}_\phi^{X_N}[f_\theta] = \lim_{N \to \infty} d\mathcal{H}_\phi[f_\theta; \psi_{\tilde{x}}^N], \tag{6}$$

*where $d\mathcal{H}_\phi[f_\theta; \psi_{\tilde{x}}^N]$ is the Gateaux derivative when $f_\theta$ is perturbed in the direction of $\psi_{\tilde{x}}^N$.*

Using the chain rule for Gateaux derivatives, we can then show that backpropagation through the entire DI-Net is convergent. These properties can be summarized in the following theorem:

**Theorem 1.** *A DI-Net permits backpropagation of its outputs with respect to its input as well as all its learnable parameters. The gradients converge under any equidistributed discretization sequence.*

### 3.5 UNIVERSAL APPROXIMATION THEOREM

Functions of bounded variation are piecewise smooth, hence they can be represented as the integral of some function. Since numerical integration can yield arbitrarily small invariance error, DI-Nets are universal approximators in the following sense:

**Theorem 2.** *For every Lipschitz continuous map $\mathcal{R} : \mathcal{F}_c \to \mathcal{F}_n$, $c, n \in \mathbb{N}$, there exists a DI-Net that approximates it to arbitrary accuracy w.r.t. a finite measure $\nu$ on $\mathcal{F}_c$. As a corollary, every Lipschitz continuous map $\mathcal{F}_c \to \mathbb{R}^n$ or $\mathbb{R}^n \to \mathcal{F}_c$ can also be approximated by some DI-Net.*

A high-level sketch of the $\mathcal{F}_1 \to \mathcal{F}_1$ case is as follows. Appendix B.2 provides a full proof, including the extension to multi-channel NFs.

---

[4] Any equidistributed sequence of points generates an equidistributed discretization sequence by truncating to the first $N$ terms, although the class of all equidistributed discretization sequences is much larger than this. Quasi-Monte Carlo sampling can efficiently generate equidistributed sequences on a wide range of domains.

1. Fix a discretization $X$ of sufficiently low discrepancy to approximate any function in $\mathcal{F}_1$ to desired accuracy. Let $N = |X|$.

2. Let $\pi$ be the projection that maps any function $f \in \mathcal{F}_1$ to $\mathbb{R}^N$ by selecting its $N$ values along the discretization. Through $\pi$, the measure $\nu$ on $\mathcal{F}_1$ induces a measure $\mu$ on $\mathbb{R}^N$.

3. We can approximate any function in $L^2(\mathbb{R}^N)$ by covering the volume under the graph of the function with almost disjoint rectangles, and then at inference time summing the heights of the rectangles at the given $\mathbb{R}^N$ input.

4. Note that a multilayer perceptron (MLP) can approximate this rectangle cover to arbitrary accuracy with sufficiently steep slopes at their boundaries (Lu et al., 2017).

5. $\hat{\mathcal{R}}^x : \pi f \mapsto \mathcal{R}[f](x)$ is in $L^2(\mathbb{R}^N)$ for each $x \in X$. We want to build a DI-Net that specifies the desired connections on $\Omega$ using element-wise products with cutoff functions and linear combinations of channels. The cutoff functions extract the input values along $X$ into separate channels, and the weights of the channels match the weights of the hypothetical MLP from step 4.

6. We repeat this construction $N$ times to specify values at each of the $N$ output points in $X$, and map all other output points to the value of the closest specified point. Then we have fully specified the desired behavior of $f \mapsto \mathcal{R}[f]$ to desired accuracy w.r.t. the measure $\nu$.

This construction points to the similarity between specifying a map between $\mathcal{F}_1 \to \mathcal{F}_1$ and specifying maps between two grids of $N$ points. Thus many of the strategies for imposing structure on how different points influence each other can inform DI-Net design. For example, if the influence should be local and translation-invariant, we can design convolutional DI-Nets. If the influence should be sparse, we can design attention-based DI-Nets. If the influence should be distributed globally among small patches, we can design transformer-like DI-Nets with tokenization. If the influence should be modulated by lower frequency patterns, we can use Fourier neural operators (Li et al., 2020).

## 4 DESIGN AND IMPLEMENTATION OF DI-NETS

DI-Nets encompass a very large family of neural networks: we have only specified the architecture as a directed acyclic graph, and DI layers include a broad variety of network layers. DI-Nets include DeepONets and neural operators (Kovachki et al., 2021b; Lu et al., 2021), which can learn general maps between function spaces but in practice are designed to solve partial differential equations. DI-Nets also encompass continuous adaptations of networks designed on discrete domains such as convolutional neural networks (CNNs). In the same way that neural fields extend signals on point clouds, meshes, grids, and graphs to a compact metric space, DI-Nets extend networks that operate on discrete signals by converting every layer to an equivalent discretizable map. We make this connection concrete in the case of CNNs below.

### 4.1 CONVOLUTIONAL DI-NETS

We describe how to extend convolutional layers and multi-scale architectures to DI-Nets here (also see Fig. 2), and discuss other layers in Appendix C, including normalization, max pooling, tokenization and attention. The resulting convolutional DI-Net can be initialized directly with the weights of a pre-trained CNN as we investigate in Appendix E.1.

**Convolution** For a measurable $S \subset \Omega$ and a polynomial basis $\{p_j\}_{j \geq 0}$ that spans $L^2(S)$, $S$ is the support of a polynomial convolutional kernel $K_\phi : \Omega \times \Omega \to \mathbb{R}$ defined by:

$$K_\phi(\boldsymbol{x}, \boldsymbol{x}') = \begin{cases} \sum_{j=1}^n \phi_j p_j(\boldsymbol{x} - \boldsymbol{x}')^j & \text{if } \boldsymbol{x} - \boldsymbol{x}' \in S \\ 0 & \text{otherwise.} \end{cases} \tag{7}$$

for some chosen $n \in \mathbb{N}$. A convolution is the linear map $\mathcal{H}_\phi : \mathcal{F}_1 \to \mathcal{F}_1$ given by:

$$\mathcal{H}_\phi[f] = \int_\Omega K_\phi(\cdot, \boldsymbol{x}') f(\boldsymbol{x}') d\boldsymbol{x}'. \tag{8}$$

An MLP convolution is defined similarly except the kernel becomes $\tilde{K}_\phi(\boldsymbol{x}, \boldsymbol{x}') = \mathrm{MLP}(\boldsymbol{x} - \boldsymbol{x}'; \phi)$ in the non-zero case. While MLP kernels are favored over polynomial kernels in many applications due to their expressive power (Wang et al., 2021), polynomial bases can be used to construct filters

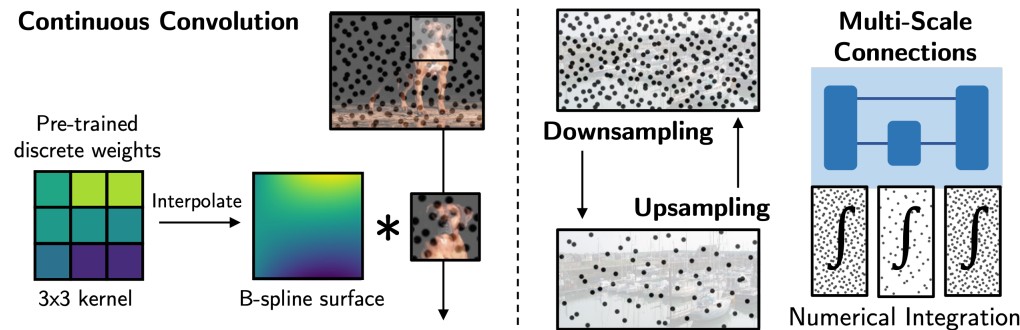

Figure 2: Convolutional DI-Nets generalize convolutional neural networks to arbitrary discretizations of the domain. Low discrepancy point sets used in quasi-Monte Carlo integration are amenable to the multi-scale structures often found in discrete networks. Convolutional DI-Nets may be initialized directly from pre-trained CNNs.

satisfying desired properties such as group equivariance (Cohen & Welling, 2016a;b), $k$-Lipschitz continuity, or boundary conditions.

The input and output discretizations of the layer can be chosen independently, allowing for padding or striding (see Appendix C). The input discretization fully determines which points on $S$ are evaluated.

**Multi-scale architectures**    Many discretizations permit multi-scale structures by subsampling the discretization, and QMC is particularly conducive to such design. Under QMC, downsampling is efficiently implemented by truncating the list of coordinates in the low discrepancy sequence to the desired number of terms, as the truncated sequence is itself low discrepancy. Similarly, upsampling can be implemented by extending the low discrepancy sequence to the desired number of terms, then performing interpolation by copying the nearest neighbor(s) or applying some (fixed or learned) kernel. Residual or skip connections can also be implemented efficiently since downsampling and upsampling are both specified with respect to the same discretization (Fig. 2 right).

## 4.2    TRAINING DI-NETS

The pipeline for training DI-Nets is similar to that for training discrete networks, except that input and/or output discretizations should be specified. When training a DI-Net classifier, the input discretization may be specified manually or sampled from a low discrepancy sequence to perform QMC integration. When training DI-Nets for dense prediction, the output discretization should be chosen to match the coordinates of the ground truth labels. Any input discretization can be chosen – in most experiments we set it equal to the output discretization. At inference time, the network can be evaluated with any output discretization (Fig. D.2), making the output in effect an NF. We outline steps for training a classifier and dense prediction DI-Net in Algorithms 2 and 3.

## 5    EXPERIMENTS

We apply convolutional DI-Nets to toy classification (NF→scalar) and dense prediction (NF→NF) tasks, and analyze its behavior under different discretizations. Our aim is not to compete with discrete networks on these tasks, but rather to demonstrate that simply deriving DI-Nets from CNNs yields a feasible class of models for discretization invariant learning on NFs, without introducing any new techniques or types of layers. Appendix D provides further experimental details. We discuss techniques that could be leveraged to design more competitive DI-Nets in Appendix F.

### 5.1    NEURAL FIELD CLASSIFICATION

We perform classification on a dataset of 8,400 NFs fit to a subset of ImageNet1k (Deng et al., 2009), with 700 samples from each of 12 superclasses (Engstrom et al., 2019). For each class we train on 500 SIRENs (Sitzmann et al., 2020b) and evaluate on 200 Gaussian Fourier feature networks (Tancik et al., 2020b).

We train DI-Nets with 2 and 4 MLP convolutional layers, as well as CNNs with similar architectures. We also train an MLP that predicts class labels from SIREN parameters, and a "non-uniform

convolution" (Jiang et al., 2019) that applies a non-uniform Fourier transform to input points (sampled with QMC) to map them to grid values, then applies a 2-layer CNN.

Each network is trained for 8K iterations with a learning rate of $10^{-3}$. In training, the CNNs sample neural fields along the $32 \times 32$ grid. DI-Nets and the non-uniform network sample 1024 points generated from a scrambled Sobol sequence (QMC discretization). We evaluate models with top-1 accuracy at the same resolution as well as at several other resolutions.

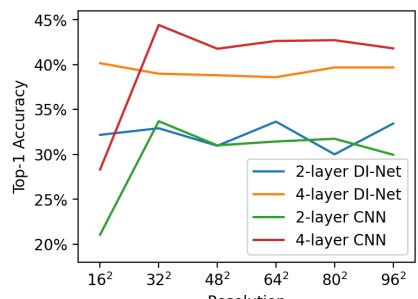

Figure 3: Classifier performance with different resolutions at test time.

The MLP and the non-uniform method significantly underperform DI-Net, with 13.9% and 28.9% accuracy respectively compared to 32.9% for our 2-layer network. At $32 \times 32$ resolution, DI-Nets somewhat underperform their CNN counterparts, and this performance gap is larger for deeper models. However, our discretization invariant model better generalizes to images of different resolutions than CNNs (Fig. 3), particularly at lower resolutions.

We next examine whether DI-Net can adapt to an entirely different type of discretization at test time. We use grid, QMC and shrunk discretizations of 1024 ($32 \times 32$) points. The Shrunk discretization shrinks a Sobol (QMC) sequence towards the center of the image (each point $x \in [-1, 1]^2$ is mapped to $x^2 \mathrm{sgn}(x)$). In image classification, the object of interest is usually centered, hence the shrunk→shrunk setting performs on par with other discretizations despite its higher discrepancy.

Table 1: Accuracy of 2-layer DI-Net under various discretizations.

| Train→Test Disc. | Accuracy |
|---|---|
| QMC→QMC | **32.9%** |
| Grid→Grid | 30.5% |
| Shrunk→Shrunk | 30.3% |
| QMC→Grid | 27.1% |
| Grid→QMC | 27.8% |
| QMC→Shrunk | 25.4% |
| Shrunk→QMC | 13.4% |

Interestingly, changing discretization type at inference time has varying impact. Usually it only slightly degrades DI-Net's accuracy (Table 1), but performance falls dramatically when shifting from high to low discrepancy discretizations (shrunk→QMC). Thus discretization invariance only provides a weak guarantee on the stability of a model's behavior, and points to the importance of training on the right discretizations to attain a network that generalizes well to other discretizations for the given task.

## 5.2 NEURAL FIELD SEGMENTATION

We perform semantic segmentation of SIRENs fit to street view images from Cityscapes (Cordts et al., 2016), grouping segmentation labels into 7 categories. We train on 2975 NFs with coarsely annotated segmentations only, and test on 500 NFs with both coarse and fine annotations (Fig. 4). We use a $48 \times 96$ grid discretization since segmentation labels are only given at pixel coordinates. We compare the performance of 3 and 5 layer DI-Nets and fully convolutional networks (FCNs), as well as a non-uniform CNN (Jiang et al., 2019). We also train a hypernetwork that learns to map each SIREN to the parameters of a new SIREN representing its segmentation.

Networks are trained for 10K iterations with a learning rate of $10^{-3}$. We evaluate each model with mean intersection over union (mIoU) and pixel-wise accuracy (PixAcc).

The hypernetwork and non-uniform CNN perform poorly compared to both FCNs and DI-Nets (Table 2). DI-Net-3 outperforms the equivalent FCN, and less often confuses features such as shadows and road markings (Fig. 4). However, the performance deteriorates when downsampling and upsampling layers are added (DI-Net-5), echoing the difficulty in scaling DI-Nets observed in classification. We suggest potential methods for remedying this in Appendix F.

Table 2: Segmentation performance on NFs fit to Cityscapes images (trained on coarse segs).

| Model Type | Coarse Segs | | Fine Segs | |
|---|---|---|---|---|
| | mIoU | PixAcc | mIoU | PixAcc |
| 3-layer FCN | 0.409 | 69.6% | 0.374 | 63.6% |
| DI-Net-3 | **0.471** | **78.5%** | **0.417** | **69.4%** |
| 5-layer FCN | **0.488** | **79.4%** | **0.436** | **72.5%** |
| DI-Net-5 | 0.443 | 77.7% | 0.394 | 68.4% |
| Hypernetwork | 0.038 | 7.9% | 0.042 | 8.3% |
| Non-uniform | 0.109 | 26.5% | 0.106 | 22.7% |

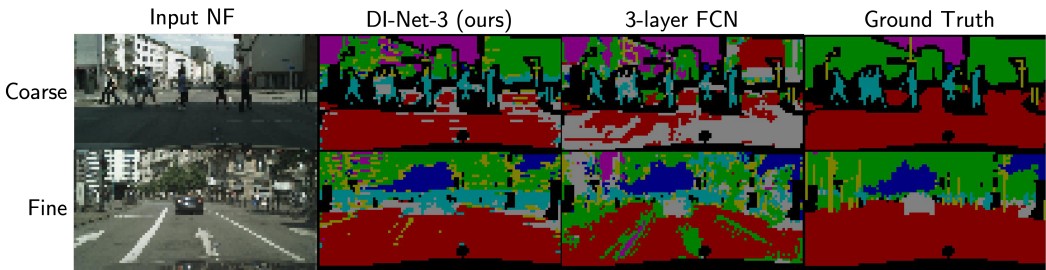

Figure 4: Cityscapes NF segmentations for models trained on coarse segmentations only. NF-Net produces NF segmentations, which can be evaluated at the subpixel level.

### 5.3 SIGNED DISTANCE FUNCTION PREDICTION

We train a convolutional DI-Net to map a field of RGBA values in 3D to its signed distance function (SDF). We create a synthetic dataset of 3D scenes with randomly colored balls embedded in 3D space, and train on RGBA-SDF pairs using a mean squared error (MSE) loss on the predicted SDF. We train with grid ($16 \times 16 \times 16$), QMC, shrunk, Monte Carlo (i.i.d. points drawn uniformly from the domain), and mixed discretizations of 4096 points. In the mixed setting, each minibatch uses one of the other four discretizations at random.

Under any fixed discretization, the convolutional DI-Net significantly outperforms the equivalent discrete network (MSE of 0.022 vs. 0.067 respectively). In Figure D.2, we illustrate our model's ability to also produce outputs that are discretized differently than the input, making DI-Net's output in effect a neural field. By changing the output discretization of the last convolutional DI-Net layer, we can evaluate the output SDF anywhere on the domain without changing the input discretization. Whereas the discrete network is forced to output predictions at the resolution it was trained on, convolutional DI-Net can produce outputs along a high-quality grid discretization given a coarse QMC discretization, even when it is only trained under QMC output discretizations.

DI-Net performs almost equally well under grid, QMC and shrunk discretizations in the in-distribution setting, but on this task it is more sensitive to out-of-distribution discretizations than the classifier in Section 5.1. While shrunk→QMC is the worst performing combination for the classifier, here it is one of the better performing combinations. DI-Net likely struggles with Monte Carlo sampling due to its high discrepancy discretizations, resulting in cases where smaller balls are entirely missed. Interestingly, the model fares worse when trained on multiple discretizations simultaneously, suggest-

Table 3: Mean squared error ($\times 10^{-2}$) of predicted SDFs under different discretizations. Top 3 settings bolded. MC=Monte Carlo.

| Train \ Test | Grid | QMC | Shrunk | MC |
|---|---|---|---|---|
| Grid | **2.18** | 2.54 | 3.77 | 3.81 |
| QMC | 3.60 | **2.01** | 2.94 | 3.92 |
| Shrunk | 3.72 | 2.88 | **2.00** | 4.30 |
| MC | 6.45 | 5.97 | 4.89 | 5.92 |
| Mixed | 4.65 | 4.41 | 3.26 | 4.09 |

ing that the network may be guided in opposing directions by different discretizations resulting in unstable training. These observations illustrate the complex task-dependent interplay between the type of discretizations observed at training time and the ability of the model to generalize to new discretizations.

## 6 CONCLUSION

DI-Net constitutes the first discretization invariant sampling approach for performing inference directly on neural fields. Motivated by discretization invariance, we designed a parameterization based on numerical integration that gives rise to strong convergence properties and a universal approximation theorem for a wide class of functions. Not only is our framework agnostic to NF parameterization, but it also extends to functions of bounded variation over a wide class of domains, and thus can be applied to many systems that process a continuous signal by querying it on a subset of its domain. We outline several directions for enhancing such models in Appendix F, which could enable them to scale to deeper architectures and tackle more challenging tasks. With the increasing popularity and diversity of neural fields as well as the emergence of tools to efficiently create large datasets of NFs, DI-Nets may become an attractive option when interoperability and discretization invariance are desired.

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

# Appendix

Appendix A provides additional background on the variation of a function and discrepancy of a point set, as well as more general forms of DI layers. Appendix B provides proofs of the Universal Approximation and Convergent Empirical Gradients theorems, as well as extensions of these properties from the single-channel case stated in the main text to multi-channel maps. Appendix C provides a detailed specification of DI-Net layers that enable DI-Nets to replicate the behavior of grid-based networks. Appendix D provides additional details of the data and architectures used in our classification experiment. Appendix E provides several analyses including properties of DI-Nets under different discretizations. Appendix F describes limitations and directions for future work.

## A  MORE DETAILS ON DISCRETIZATION INVARIANCE

### A.1  KOKSMA–HLAWKA INEQUALITY AND LOW DISCREPANCY SEQUENCES

Recall that a function $f \in L^2(\Omega)$ satisfies a Koksma–Hlawka inequality if for any point set $X \subset \Omega$,

$$\left| \frac{1}{|X|} \sum_{x' \in X} f(x') - \int_\Omega f(x) \, dx \right| \leq V(f) \, D(X), \tag{9}$$

for normalized measure $dx$, some notion of variation $V$ of the function and some notion of discrepancy $D$ of the point set. The classical inequality gives a tight error bound for functions of bounded variation in the sense of Hardy-Krause (BVHK), a generalization of bounded variation to multivariate functions on $[0,1]^d$ which has bounded variation in each variable. Specifically, the variation is defined as:

$$V_{HK}(f) = \sum_{\alpha \in \{0,1\}^d} \int_{[0,1]^{|\alpha|}} \left| \frac{\partial^\alpha}{\partial x^\alpha} f(x_\alpha) \right| dx, \tag{10}$$

with $\{0,1\}^d$ the multi-indices and $x_\alpha \in [0,1]^d$ such that $x_{\alpha,j} = x_j$ if $j \in \alpha$ and $x_{\alpha,j} = 1$ otherwise. The classical inequality also uses the star discrepancy of the point set $X$, given by:

$$D^*(X) = \sup_{I \in J} \left| \frac{1}{|X|} \sum_{x' \in X} \mathbb{1}_I(x_j) - \lambda(I) \right|, \tag{11}$$

where $J$ is the set of $d$-dimensional intervals that include the origin, and $\lambda$ the Lebesgue measure.

A point set is called low discrepancy if its discrepancy is on the order of $O((\ln N)^d/N)$. Quasi-Monte Carlo calculates the sample mean using a low discrepancy sequence (see Fig. A.1 for examples in 2D), as opposed to the i.i.d. point set generated by standard Monte Carlo, which will generally be high discrepancy. Because the Koksma–Hlawka inequality is sharp, when estimating the integral of a BVHK function on $[0,1]^d$, the error of the QMC approximation decays as $O((\ln N)^d/N)$, in contrast to the error of the standard Monte Carlo approximation that decays as $O(N^{-1/2})$ (Caflisch, 1998).

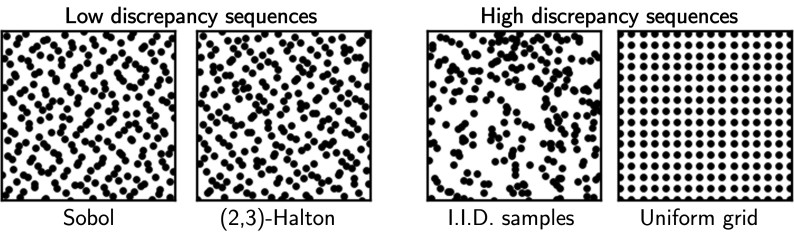

Figure A.1: Examples of low and high discrepancy sequences in 2D.

However, BVHK is a rather restrictive class of functions defined on $[0,1]^d$ that excludes all functions with discontinuities. Brandolini et al. (2013) extended the Koksma–Hlawka inequality to two classes of functions defined below:

**Piecewise smooth functions**    Let $f$ be a smooth function on $[0,1]^d$ and $\Omega$ a Borel subset of $[0,1]^d$. Then $f|_\Omega$ is a piecewise smooth function with the Koksma–Hlawka inequality given by variation

$$V(f) = \sum_{\alpha \in \{0,1\}^d} 2^{d-|\alpha|} \int_{[0,1]^d} \left| \frac{\partial^\alpha}{\partial x^\alpha} f(x) \right| dx, \tag{12}$$

and discrepancy:

$$D(X) = 2^d \sup_{I \subseteq [0,1]^d} \left| \frac{1}{|X|} \sum_{x' \in X} \mathbb{1}_{\Omega \cap I}(x_j) - \lambda(\Omega \cap I) \right|. \tag{13}$$

**$W^{d,1}$ functions on manifolds**    Let $\mathscr{M}$ be a smooth compact $d$-dimensional manifold with normalized measure $dx$. Given local charts $\{\phi_k\}_{k=1}^K$, $\phi_k : [0,1]^d \to \mathscr{M}$, the variation of a function $f \in W^{d,1}(\mathscr{M})$ is characterized as:

$$V(f) = c \sum_{k=1}^K \sum_{|\alpha| \leq n} \int_{[0,1]^d} \left| \frac{\partial^\alpha}{\partial x^\alpha} (\psi_k(\phi_k(x)) f(\phi_k(x))) \right| dx, \tag{14}$$

with $\{\psi_k\}_{k=1}^K$ a smooth partition of unity subordinate to the charts, and constant $c > 0$ that depends on the charts but not on $f$. Defining the set of intervals in $\mathscr{M}$ as $J = \{U : U = \phi_k(I)$ for some $k$ and $I \subseteq [0,1]^d\}$, with measure $\mu(U) = \lambda(I)$, the discrepancy of a point set $Y = \{y_j\}_{y=1}^N$ on $\mathscr{M}$ is:

$$D(Y) = \sup_{U \in J} \left| \frac{1}{|X|} \sum_{x' \in X} \mathbb{1}_U(y_j) - \mu(U) \right|. \tag{15}$$

**Neural fields**    We define the variation of a neural field as the sum of the variations of each channel.

**Note**: The notion of discrepancy is not limited to the Lebesgue measure. The existence of low discrepancy point sets has been proven for non-negative, normalized Borel measures on $[0,1]^d$ due to Aistleitner & Dick (2013). An extension of our framework to non-uniform measures is a promising direction for future work (see Appendix F).

## A.2    MORE GENERAL FORMS OF DI LAYERS

Recall that we defined DI layers as having the form $\mathcal{H}_\phi[f] = \int_\Omega H_\phi[f](x)dx$ for neural fields $f$ (we drop $\theta$ here for readability). In the case where $H_\phi[f] : \Omega \to \mathbb{R}^n$, i.e. $\mathcal{H}_\phi$ is a layer that maps neural fields to vectors, we permit layers of the following more general form:

$$\mathcal{H}_\phi[f] = \int_\Omega h_\phi(x, f(x), \dots, D^\alpha f(x))d\mu(x), \tag{16}$$

for weak derivatives up to order $|\alpha|$ taken with respect to each channel. $D^\alpha f = \frac{\partial^{|\alpha|} f}{\partial x_1^{\alpha_1} \dots \partial x_n^{\alpha_n}}$ for multi-index $\alpha$. The dependence of $h$ on weak derivatives up to order $k = |\alpha|$ requires that the weak derivatives are integrable, i.e., $f$ is in the Sobolev space $W^{k,2}(\Omega)$, and that $h_\phi$ is Gateaux differentiable w.r.t. these weak derivatives. Note that a non-uniform measure $\mu$ changes the discrepancy of sampled sequences.

In the NF-valued case, we can similarly have:

$$\mathcal{H}_\phi[f](x') = \int_\Omega H_\phi[f](x, x')d\mu(x) \tag{17}$$

$$= \int_\Omega h_\phi(x, f(x), \dots, D^\alpha f(x), x', f(x'), \dots, D^\alpha f(x'))d\mu(x), \tag{18}$$

where we require $H_\phi[f](\cdot, x') \in \mathcal{F}_n$. Both of our key theoretical results (convergence of discretized gradients and universal approximation) apply to this general form. Gateaux differentiability of $h_\phi$ allows us to apply the same proof in Appendix B.1 to the derivatives. Since allowing layers to depend on weak derivatives results in an even more expressive class of DI-Nets, the universal approximation theorem still holds.

# B  PROOFS

## B.1  PROOF OF THEOREM 1 (CONVERGENCE OF DISCRETIZED GRADIENTS)

*A DI-Net permits backpropagation of its outputs with respect to its input as well as all its learnable parameters. Under an equidistributed discretization sequence, the gradients of each layer converge to the appropriate derivative under the measure on $\Omega$.*

We note that this property automatically holds if the layer does not perform numerical integration. This includes layers which take $\mathbb{R}^n$ as input, as well as point-wise transformations. Then the (sub)derivatives with respect to inputs and parameters need only be well-defined at each point of the output in order to enable backpropagation.

Choose an equidistributed discretization sequence $\{X_N\}_{N\in\mathbb{N}}$ on $\Omega$. We consider a DI layer $\mathcal{H}_\phi$ which takes an NF $f$ (we drop dependence on $\theta$ for readability) as input and may output a vector or NF.

$$\mathcal{H}_\phi[f] = \int_\Omega H_\phi[f](x)dx. \tag{19}$$

Recall we write the estimate under $X_N$ as:

$$\hat{\mathcal{H}}_\phi^N[f] = \frac{1}{|X_N|} \sum_{x\in X_N} H_\phi[f](x). \tag{20}$$

and call its derivatives the discretized derivatives of $\mathcal{H}_\phi[f]$. We are interested in proving the convergence of the discretized gradients of $\mathcal{H}_\phi[f]$ with respect to its input $f$ as well as the parameters $\phi$ of the layer.

**Definition 2.** *For given discretization $X$, the projection $\pi : f \mapsto \mathbf{f}$ is a quotient map $L^2(\Omega) \to L^2(\Omega)/\sim$ under the equivalence relation $f \sim g$ iff $f(x) = g(x)$ for all $x \in X$.*

Thus we can write $\pi f = \{f(x)\}_{x\in X}$ when the discretization is clear from the context.

**Lemma 1.** *For any DI-Net layer $\mathcal{H}_\phi$ which takes an NF $f \in \mathcal{F}_m$ as input, the discretized gradient of $\mathcal{H}_\phi[f]$ w.r.t. its parameters $\phi$ is convergent in $N$:*

$$\lim_{N\to\infty} \left| \nabla_\phi \hat{\mathcal{H}}_\phi^N[f] \right| < \infty. \tag{21}$$

*Additionally, the discretized gradient of $\mathcal{H}_\phi[f]$ w.r.t. its discretized input $\pi f$ is convergent in $N$:*

$$\lim_{N\to\infty} \left| \nabla_{\pi f} \hat{\mathcal{H}}_\phi^N[f] \right| < \infty. \tag{22}$$

*Proof.* **NF to Vector: gradients w.r.t. parameters**

Consider the case of a layer $\mathcal{H}_\phi : \mathcal{F}_m \to \mathbb{R}^n$. If $\phi = (\phi_1, \ldots, \phi_K)$, then denote $\phi + \tau e_k = (\phi_1, \ldots, \phi_{k-1}, \phi_k + \tau, \phi_{k+1}, \ldots, \phi_K)$.

$$\lim_{N\to\infty} \frac{\partial}{\partial \phi_k} \hat{\mathcal{H}}_\phi^N[f] = \lim_{N\to\infty} \frac{\partial}{\partial \phi_k} \left( \frac{1}{|X_N|} \sum_{x\in X_N} H_\phi[f](x) \right) \tag{23}$$

$$= \lim_{N\to\infty} \lim_{\tau\to0} \left( \frac{1}{\tau N} \sum_{j=1}^N H_{\phi+\tau e_k}[f](x) - H_\phi[f](x) \right) \tag{24}$$

$$= \lim_{\tau\to0} \frac{1}{\tau} \int_\Omega H_{\phi+\tau e_k}[f](x) - H_\phi[f](x)\, dx \tag{25}$$

$$= \lim_{\tau\to0} \frac{\mathcal{H}_{\phi+\tau e_k}[f] - \mathcal{H}_\phi[f]}{\tau} \tag{26}$$

$$= \frac{\partial}{\partial \phi_k} \mathcal{H}_\phi[f], \tag{27}$$

where (25) follows by (16) and the Moore-Osgood theorem. Thus the discretized gradient converges to the Jacobian of $\mathcal{H}_\phi$ w.r.t. each parameter, which is finite by differentiability and boundedness of $h_\phi$.

**NF to NF: gradients w.r.t. parameters**

From (17), we stated that a DI-Net layer $\bar{\mathcal{H}}_{\phi'} : L^2(\Omega) \to L^2(\Omega)$ can be expressed as:

$$\bar{\mathcal{H}}_{\phi'}[f](x') = \int_\Omega \bar{h}(x, f(x), \dots, D^\alpha f(x), x', f(x'), \dots, D^\alpha f(x'); \phi')dx. \tag{28}$$

We can in fact follow the same steps as the NF to Vector case above, to arrive at:

$$\lim_{N \to \infty} \frac{\partial}{\partial \phi'_k} \left( \frac{1}{|X_N|} \sum_{x \in X_N} \bar{H}_{\phi'}[f] \right) = \frac{\partial}{\partial \phi'_k} \bar{\mathcal{H}}_{\phi'}[f], \tag{29}$$

with equality at each point $x' \in \Omega$ and channel of the output NF.

**NF input: gradients w.r.t. inputs**

Here we combine the NF to vector and NF to NF cases for brevity. For fixed $\tilde{x} \in \Omega$, the discretized derivative of $\mathcal{H}_\phi$ w.r.t. $f(\tilde{x})$ can be written:

$$\frac{\partial}{\partial f(\tilde{x})} \hat{\mathcal{H}}_\phi^N[f] = \frac{\partial}{\partial f(\tilde{x})} \left( \frac{1}{|X_N|} \sum_{x \in X_N} H_\phi[f](x) \right) \tag{30}$$

$$= \lim_{\tau \to 0} \frac{1}{\tau |X_N|} \sum_{x \in X_N} H_\phi[f + \tau \psi_{\tilde{x}}^N](x) - H_\phi[f](x), \tag{31}$$

where $\psi_{\tilde{x}}^N$ is any function in $W^{|\alpha|,1}(\Omega)$ that is 1 at $\tilde{x}$ and 0 on $X_N \backslash \{\tilde{x}\}$, and whose derivatives are 0 on $X_N$. As an example, take the bump function which vanishes outside a small neighborhood of $\tilde{x}$ and smoothly ramps to 1 on a smaller neighborhood of $\tilde{x}$, making its weak derivative 0 at $\tilde{x}$.

By (3) we know that the sequences $\left\| \hat{\mathcal{H}}_\phi^N[f] - \mathcal{H}_\phi[f] \right\|$ and $\left\| \hat{\mathcal{H}}_\phi^N[f + \tau \psi_{\tilde{x}}^N] - \mathcal{H}_\phi[f + \tau \psi_{\tilde{x}}^N] \right\|$ converge uniformly in $N$ to 0 for any $\tau > 0$, where we can use the $\ell_1$ norm for vector outputs or the $L^1$ norm for NF outputs. So for any $\epsilon > 0$ and any $\tau > 0$, we can choose $N_0$ large enough such that for any $N > N_0$:

$$\left\| \hat{\mathcal{H}}_\phi^N[f + \tau \psi_{\tilde{x}}^N] - \mathcal{H}_\phi[f + \tau \psi_{\tilde{x}}^N] \right\| < \frac{\epsilon}{2}, \tag{32}$$

and

$$\left\| \hat{\mathcal{H}}_\phi^N[f] - \mathcal{H}_\phi[f] \right\| < \frac{\epsilon}{2}. \tag{33}$$

Then,

$$\left\| \hat{\mathcal{H}}_\phi^N[f + \tau \psi_{\tilde{x}}^N] - \mathcal{H}_\phi[f + \tau \psi_{\tilde{x}}^N] \right\| + \left\| \hat{\mathcal{H}}_\phi^N[f] - \mathcal{H}_\phi[f] \right\| < \epsilon, \tag{34}$$

by the triangle inequality,

$$\left\| (\hat{\mathcal{H}}_\phi^N[f + \tau \psi_{\tilde{x}}^N] - \mathcal{H}_\phi[f + \tau \psi_{\tilde{x}}^N]) - (\hat{\mathcal{H}}_\phi^N[f] - \mathcal{H}_\phi[f]) \right\| < \epsilon \tag{35}$$

$$\left\| \left( \hat{\mathcal{H}}_\phi^N[f + \tau \psi_{\tilde{x}}^N] - \hat{\mathcal{H}}_\phi^N[f] \right) - \left( \mathcal{H}_\phi[f + \tau \psi_{\tilde{x}}^N] - \mathcal{H}_\phi[f] \right) \right\| < \epsilon, \tag{36}$$

hence $\left\| \left( \hat{\mathcal{H}}_\phi^N[f + \tau \psi_{\tilde{x}}^N] - \hat{\mathcal{H}}_\phi^N[f] \right) - \left( \mathcal{H}_\phi[f + \tau \psi_{\tilde{x}}^N] - \mathcal{H}_\phi[f] \right) \right\|$ converges uniformly to 0. Since the distance between two vectors is 0 iff they are the same, we can write:

$$\lim_{N \to \infty} \hat{\mathcal{H}}_\phi^N[f + \tau \psi_{\tilde{x}}^N] - \hat{\mathcal{H}}_\phi^N[f] = \lim_{N \to \infty} \mathcal{H}_\phi[f + \tau \psi_{\tilde{x}}^N] - \mathcal{H}_\phi[f] \tag{37}$$

$$\lim_{\tau \to 0} \frac{1}{\tau} \lim_{N \to \infty} \left( \hat{\mathcal{H}}_\phi^N[f + \tau \psi_{\tilde{x}}^N] - \hat{\mathcal{H}}_\phi^N[f] \right) = \lim_{\tau \to 0} \frac{1}{\tau} \lim_{N \to \infty} \left( \mathcal{H}_\phi[f + \tau \psi_{\tilde{x}}^N] - \mathcal{H}_\phi[f] \right). \tag{38}$$

By the Moore-Osgood theorem,

$$\lim_{N \to \infty} \lim_{\tau \to 0} \frac{1}{\tau} \left( \hat{\mathcal{H}}_\phi^N[f + \tau \psi_{\tilde{x}}^N] - \hat{\mathcal{H}}_\phi^N[f] \right) = \lim_{N \to \infty} \lim_{\tau \to 0} \frac{1}{\tau} \left( \mathcal{H}_\phi[f + \tau \psi_{\tilde{x}}^N] - \mathcal{H}_\phi[f] \right) \quad (39)$$

$$\lim_{N \to \infty} \frac{\partial}{\partial f(\tilde{x})} \hat{\mathcal{H}}_\phi^N[f] = \lim_{N \to \infty} d\mathcal{H}_\phi[f; \psi_{\tilde{x}}^N]. \quad (40)$$

Since $h_\phi$ is Gateaux differentiable and bounded, $H_\phi$ is also Gateaux differentiable for $f$ of bounded variation, hence the limit on the right hand side is finite.

For each discretization $X_N$, choose a sequence of bump functions around each $x \in X_N$, $\{\psi_x^N\}_{x \in X_N}$. An example of such a family is the (appropriately designed) partitions of unity with $|X_N|$ elements.

Then the discretized gradient w.r.t. $\pi f$ converges to the limit of the Gateaux derivatives of $\mathcal{H}_\phi$ w.r.t. the bump function sequence as $N \to \infty$. $\qquad \square$

**Lemma 2.** *Chained discretized derivatives converge to the chained Gateaux derivatives.*

*Proof.* Consider a two-layer DI-Net with NF input $f \mapsto (\mathcal{H}_\theta \circ \mathcal{H}_\phi)[f]$. For the case of derivatives w.r.t. the input, we would like to show the analogue of (40):

$$\lim_{N \to \infty} \frac{\partial}{\partial f(\tilde{x})} \left( \hat{\mathcal{H}}_\theta^N \circ \hat{\mathcal{H}}_\phi^N \right)[f] = \lim_{N \to \infty} d\left( \mathcal{H}_\theta \circ \mathcal{H}_\phi \right)[f; \psi_{\tilde{x}}^N], \quad (41)$$

where the bump function $\psi_{\tilde{x}}^N$ is defined similarly (1 at $\tilde{x}$ and 0 at each $x \neq \tilde{x}$).

$$\frac{\partial}{\partial f(\tilde{x})} \left( \hat{\mathcal{H}}_\theta^N \circ \hat{\mathcal{H}}_\phi^N \right)[f] = \frac{\partial}{\partial f(x)} \left( H_\theta \left[ \hat{\mathcal{H}}_\phi^N[f] \right](x) \right) \quad (42)$$

$$= \lim_{\tau \to 0} \frac{1}{\tau} \left( H_\theta \left[ \hat{\mathcal{H}}_\phi^N[f + \tau \psi_{\tilde{x}}^N] \right](x) - H_\theta \left[ \hat{\mathcal{H}}_\phi^N[f] \right](x) \right) \quad (43)$$

as in (31).

By (3) we know $\left\| \mathcal{H}_\theta \left[ \hat{\mathcal{H}}_\phi^N[f + \tau \psi_{\tilde{x}}^N] \right] - \hat{\mathcal{H}}_\theta^N \left[ \hat{\mathcal{H}}_\phi^N[f + \tau \psi_{\tilde{x}}^N] \right] \right\|$ converges to 0 in $N$ for all $\tau > 0$ (where we can use the $\ell_1$ norm for vector outputs or the $L^1$ norm for NF outputs), as does $\left\| \mathcal{H}_\theta \left[ \hat{\mathcal{H}}_\phi^N[f] \right] - \hat{\mathcal{H}}_\theta^N \left[ \hat{\mathcal{H}}_\phi^N[f] \right] \right\|_{L^1}$. Reasoning as in (32)-(39), we have:

$$\lim_{N \to \infty} \lim_{\tau \to 0} \frac{1}{\tau} \left( \hat{\mathcal{H}}_\theta^N \left[ \hat{\mathcal{H}}_\phi^N[f + \tau \psi_{\tilde{x}}^N] \right] - \hat{\mathcal{H}}_\theta^N \left[ \hat{\mathcal{H}}_\phi^N[f] \right] \right) \quad (44)$$

$$= \lim_{\tau \to 0} \frac{1}{\tau} \lim_{N \to \infty} \left( \mathcal{H}_\theta \left[ \hat{\mathcal{H}}_\phi^N[f + \tau \psi_{\tilde{x}}^N] \right] - \mathcal{H}_\theta \left[ \hat{\mathcal{H}}_\phi^N[f] \right] \right) \quad (45)$$

$$= \lim_{N \to \infty} \lim_{\tau \to 0} \frac{1}{\tau} \left( \mathcal{H}_\theta \left[ \hat{\mathcal{H}}_\phi^N[f + \tau \psi_{\tilde{x}}^N] \right] - \mathcal{H}_\theta \left[ \hat{\mathcal{H}}_\phi^N[f] \right] \right) \quad (46)$$

Note that

$$d\mathcal{H}_\phi[f; \psi_{\tilde{x}}^N] = \frac{1}{\tau} \left( \mathcal{H}_\phi[f + \tau \psi_{\tilde{x}}^N] - \mathcal{H}_\phi[f] + o(\tau) \right) \quad (47)$$

$$\mathcal{H}_\phi[f + \tau \psi_{\tilde{x}}^N] = \mathcal{H}_\phi[f] + \tau d\mathcal{H}_\phi[f; \psi_{\tilde{x}}^N] + o(\tau). \quad (48)$$

Then we complete the equality in (41) as follows:

$$\text{LHS} = \lim_{N \to \infty} \frac{\partial}{\partial f(\tilde{x})} \left( \hat{\mathcal{H}}_\theta^N \circ \hat{\mathcal{H}}_\phi^N \right) [f] \tag{49}$$

$$= \lim_{N \to \infty} \lim_{\tau \to 0} \frac{1}{\tau} \left( \mathcal{H}_\theta \left[ \hat{\mathcal{H}}_\phi^N [f + \tau \psi_{\tilde{x}}^N] \right] - \mathcal{H}_\theta \left[ \hat{\mathcal{H}}_\phi^N [f] \right] \right) \tag{50}$$

$$= \lim_{N \to \infty} \lim_{\tau \to 0} \frac{1}{\tau} \left( \mathcal{H}_\theta \left[ \mathcal{H}_\phi[f] + \tau d\mathcal{H}_\phi[f; \psi_{\tilde{x}}^N] \right] - \mathcal{H}_\theta \left[ \mathcal{H}_\phi[f] \right] \right) \tag{51}$$

$$= \lim_{N \to \infty} d\mathcal{H}_\theta \left[ \mathcal{H}_\phi[f]; d\mathcal{H}_\phi[f; \psi_{\tilde{x}}^N] \right] \tag{52}$$

$$= \lim_{N \to \infty} d \left( \mathcal{H}_\theta \circ \mathcal{H}_\phi \right) [f; \psi_{\tilde{x}}^N] \tag{53}$$

$$= \text{RHS}, \tag{54}$$

by the chain rule for Gateaux derivatives.

The case of derivatives w.r.t. parameters is straightforward. In the same way we used (32)-(39) to obtain (46), we have:

$$\lim_{N \to \infty} \frac{\partial}{\partial \phi_k} (\hat{\mathcal{H}}_\theta^N \circ \hat{\mathcal{H}}_\phi^N)[f] = \lim_{N \to \infty} \lim_{\tau \to 0} \frac{1}{\tau} \left( \hat{\mathcal{H}}_\theta^N \left[ \hat{\mathcal{H}}_{\phi + \tau e_k}^N [f] \right] - \hat{\mathcal{H}}_\theta^N \left[ \hat{\mathcal{H}}_\phi^N [f] \right] \right) \tag{55}$$

$$= \lim_{\tau \to 0} \frac{1}{\tau} \left( \mathcal{H}_\theta [\mathcal{H}_{\phi + \tau e_k}[f]] - \mathcal{H}_\theta [\mathcal{H}_\phi[f]] \right) \tag{56}$$

$$= \frac{\partial}{\partial \phi_k} (\mathcal{H}_\theta \circ \mathcal{H}_\phi)[f], \tag{57}$$

By induction, the chained derivatives converge for an arbitrary number of layers. $\qquad \square$

Since the properties of DI-Net layers extend to loss functions on DI-Nets, we can treat a loss function similarly to a layer, and write:

$$\mathcal{L}_{g'}[g] = \int_\Omega L[g, g'](x) dx \tag{58}$$

$$\hat{\mathcal{L}}_{g'}^N[g] = \frac{1}{|X_N|} \sum_{x \in X_N} L[g, g'](x) \tag{59}$$

$$\lim_{N \to \infty} \frac{\partial}{\partial f(\tilde{x})} \left( \hat{\mathcal{L}}_{g'}^N \circ \hat{\mathcal{H}}_\theta^N \circ \hat{\mathcal{H}}_\phi^N \right) [f] = \lim_{N \to \infty} d \left( \mathcal{L}_{g'} \circ \mathcal{H}_\theta \circ \mathcal{H}_\phi \right) [f; \psi_{\tilde{x}}^N], \tag{60}$$

where $g'$ can be some other input to the loss function such as ground truth labels. Thus, we can state the following result:

**Corollary 1.** *The gradients of a DI-Net's loss function w.r.t. its inputs and all its parameters are convergent under an equidistributed discretization sequence.*

### B.2   Proof of Theorem 2 (Universal Approximation Theorem)

**Note**: By our definition of $\mathcal{F}_c$ (Section 3.1), there exists $V^*$ such that every $f \in \mathcal{F}_1$ satisfies a Koksma–Hlawka inequality (3) with $V(|f|) < V^*$.

$\mathcal{F}_1$ is bounded in $L^1$ norm since all their functions are compactly supported and bounded.

Consider a Lipschitz continuous map $\mathcal{R} : \mathcal{F}_1 \to \mathcal{F}_1$ such that $d(\mathcal{R}[f], \mathcal{R}[g])_{L^1} \le M_0 d(f, g)_{L^1}$ for some constant $M_0$ and all $f, g \in \mathcal{F}_1$. Let $M = \max\{M_0, 1\}$.

Fix a discretization $X \subset \Omega$ with discrepancy $D(X) = \frac{\epsilon}{12(M+2)V^*}$. By (3) this yields:

$$\left| \frac{1}{|X|} \sum_{x' \in X} f(x') - \int_\Omega f(x) \, dx \right| \le \frac{\epsilon}{12(M+2)}, \tag{61}$$

for all $f \in \mathcal{F}_1$. Let $N$ be the number of points in $X$.

Define the equivalence relation $\sim$ and projection $\pi$ as in Definition 2. $L^2(\Omega)/\sim$ is isomorphic to $\mathbb{R}^{|X|}$, and thus can be given the normalized $\ell^1$ norm:

$$\|\pi f\|_{\ell^1} = \frac{1}{|X|} \sum_{x' \in X} |f(x')|. \tag{62}$$

**Definition 3.** *Denote the preimage of $\pi$ as $\boldsymbol{\pi}^{-1} : \mathbf{f}' \mapsto \{f' \in \mathcal{F}_1 : \pi f' = \mathbf{f}'\}$. Invoking the axiom of choice, define the inverse projection $\pi^{-1} : \pi\mathcal{F}_1 \to \mathcal{F}_1$ by a choice function over the sets $\boldsymbol{\pi}^{-1}(\pi\mathcal{F}_1)$.*

Note that this inverse projection corresponds to some way of interpolating the $N$ sample points such that the output is in $\mathcal{F}_1$. Although our definition implies the existence of such an interpolator, we leave its specification as an open problem. Since $\Omega$ only permits discontinuities along a fixed Borel subset of $[0,1]^d$, these boundaries can be specified *a priori* in the interpolator. Since all functions in $\mathcal{F}_1$ are bounded and continuous outside this set, the interpolator can be represented by a bounded continuous map, hence it is expressible by a DI-Net layer.

**Definition 4.** *$\pi$ generates a $\sigma$-algebra on $\mathcal{F}_1$ given by $\mathscr{A} = \{\boldsymbol{\pi}^{-1}(S) : S \in \mathscr{L}\}$, with $\mathscr{L}$ the $\sigma$-algebra of Lebesgue measurable sets on $\mathbb{R}^N$. Because this $\sigma$-algebra depends on $\epsilon$ and the Lipschitz constant of $\mathcal{R}$ via the point set's discrepancy, we may write it as $\mathscr{A}_{\epsilon,\mathcal{R}}$.*

In this formulation, we let the tolerance $\epsilon$ and the Lipschitz constant of $\mathcal{R}$ dictate what subsets of $\mathcal{F}_1$ are measurable, and thus which measures on $\mathcal{F}_1$ are permitted. However, if the desired measure $\nu$ is more fine-grained than what is permitted by $\mathscr{A}_{\epsilon,\mathcal{R}}$, then it is $\nu$ that should determine the number of sample points $N$, rather than $\epsilon$ or $\mathcal{R}$.

We now state the following lemmas which will be used to prove our universal approximation theorem.

**Lemma 3.** *There is a map $\tilde{\mathcal{R}} : \pi\mathcal{F}_1 \to \pi\mathcal{F}_1$ such that*

$$\int_\Omega \left| \mathcal{R}[f](x) - \pi^{-1} \circ \tilde{\mathcal{R}} \circ \pi[f](x) \right| dx = \frac{\epsilon}{6}. \tag{63}$$

*Proof.* Let $g(x) = |f(x)|$ for $f \in \mathcal{F}_1$. Because (61) applies to $g(x)$, we have:

$$\left| \frac{1}{|X|} \sum_{x' \in X} g(x') - \int_\Omega g(x)\, dx \right| \leq \frac{\epsilon}{12(M+2)} \tag{64}$$

$$\left| \|\pi f\|_{\ell^1} - \|f\|_{L^1} \right| \leq \frac{\epsilon}{12(M+2)}. \tag{65}$$

Eqn. (65) also implies that for any $\mathbf{f} \in \pi\mathcal{F}_1$, we have:

$$\left| \|\mathbf{f}\|_{\ell^1} - \|\pi^{-1}\mathbf{f}\|_{L^1} \right| \leq \frac{\epsilon}{12(M+2)}. \tag{66}$$

Combining (65) and (66), we obtain

$$\left| \|f\|_{L^1} - \|\pi^{-1} \circ \pi[f]\|_{L^1} \right| \leq \frac{\epsilon}{6(M+2)}. \tag{67}$$

By the triangle inequality and applying $\mathcal{R}$:

$$\int_\Omega \left| \mathcal{R}[f](x) - \pi^{-1} \circ \pi \circ \mathcal{R}[f](x) \right| dx \leq \frac{\epsilon}{6(M+2)}. \tag{68}$$

For any $f, g \in \mathcal{F}_1$ such that $\pi f = \pi g$, (65) tells us that $d(f,g)_{L^1}$ is at most $\epsilon/6(M+2)$. Recall $M$ was defined such that $d(\mathcal{R}[f], \mathcal{R}[g])_{L^1} \leq M d(f,g)_{L^1}$ for any $\mathcal{R}$.

$$d(\pi \circ \mathcal{R}[f], \pi \circ \mathcal{R}[g])_{L^1} \leq \frac{M\epsilon}{6(M+2)} + \frac{\epsilon}{6(M+2)} \tag{69}$$

$$= \frac{(M+1)}{(M+2)} \frac{\epsilon}{6} \tag{70}$$

So defining:

$$\tilde{\mathcal{R}} = \arg\min_{\mathcal{H}} d(\mathcal{H} \circ \pi[f], \pi \circ \mathcal{R}[f])_{\ell^1}, \tag{71}$$

we have

$$\left| \tilde{\mathcal{R}} \circ \pi[f] - \pi \circ \mathcal{R}[f] \right| \leq \frac{(M+1)}{(M+2)} \frac{\epsilon}{6}. \tag{72}$$

Then by (68),

$$\int_\Omega \left| \mathcal{R}[f](x) - \pi^{-1} \circ \tilde{\mathcal{R}} \circ \pi[f](x) \right| dx \leq \frac{\epsilon}{6(M+2)} + \frac{(M+1)}{(M+2)} \frac{\epsilon}{6} \tag{73}$$

$$= \frac{\epsilon}{6}. \tag{74}$$

$\square$

**Lemma 4.** *Consider the extension of $\tilde{\mathcal{R}}$ to $\mathbb{R}^N \to \mathbb{R}^N$ in which each component of the output has the form:*

$$\tilde{\mathcal{R}}_j(\mathbf{f}) = \begin{cases} \mathcal{R}[\pi^{-1}\mathbf{f}](x) & \text{if } \mathbf{f} \in \pi\mathcal{F}_1 \\ 0 & \text{otherwise.} \end{cases} \tag{75}$$

*Then any finite measure $\nu$ on the measurable space $(\mathcal{F}_1, \mathscr{A})$ induces a finite measure $\mu$ on $(\mathbb{R}^N, \mathscr{L})$, and $\int_{\mathbb{R}^N} |\tilde{\mathcal{R}}_j(\mathbf{f})| \mu(d\mathbf{f}) < \infty$ for each $j$.*

*Proof.* Since the $\sigma$-algebra $\mathscr{A}$ on $\mathcal{F}_1$ is generated by $\pi$, the measure $\mu : \mu(\pi S) = \nu(S)$ for all $S \in \mathscr{A}$ is finite and defined w.r.t. the Lebesgue measurable sets on $\pi\mathcal{F}_1$. Since $\pi\mathcal{F}_1$ can be identified with a measurable subset of $\mathbb{R}^N$, $\mu$ can be naturally extended to $\mathbb{R}^N$. Doing so makes it absolutely continuous w.r.t. the Lebesgue measure on $\mathbb{R}^N$.

To show $\tilde{\mathcal{R}}_j(\mathbf{f})$ is integrable, it is sufficient to show it is bounded and compactly supported.

$\mathcal{F}_1$ is bounded in the $L^1$ norm. Thus by (65), $\pi\mathcal{F}_1$ is bounded in the normalized $\ell_1$ norm. The $\ell_1$ norm in $\mathbb{R}^N$ is strongly equivalent to the uniform norm, so there is some compact set $[-c, c]^N$, $c > 0$ for which the extension of $\pi\mathcal{F}_1$ to $\mathbb{R}^N$ vanishes, so $\text{supp}(\tilde{\mathcal{R}}_j(\mathbf{f})) \subseteq [-c, c]^N$.

Similarly, $\pi\mathcal{F}_1$ is bounded in the $\ell^1$ norm, hence there exists $c'$ such that $\tilde{\mathcal{R}}_j < c'$ for all $j$. $\square$

**Lemma 5.** *For any finite measure $\mu$ absolutely continuous w.r.t. the Lebesgue measure on $\mathbb{R}^n$, $J \in L^1(\mu)$ and $\epsilon > 0$, there is a network $\mathcal{K}$ such that:*

$$\int_{\mathbb{R}^n} |J(\mathbf{f}) - \mathcal{K}(\mathbf{f})| \mu(d\mathbf{f}) < \frac{\epsilon}{2}. \tag{76}$$

*Proof.* The following construction is adapted from Lu et al. (2017). Since $J$ is integrable, there is a cube $E = [-c, c]^n$ such that:

$$\int_{\mathbb{R}^n \setminus E} |J(\mathbf{f})| \mu(d\mathbf{f}) < \frac{\epsilon}{8} \tag{77}$$

$$\|J - \mathbb{1}_E J\|_1 < \frac{\epsilon}{8}. \tag{78}$$

*Case 1: J is non-negative on all of $\mathbb{R}^n$*

Define the set under the graph of $J|_E$:

$$G_{E,J} \triangleq \{(\mathbf{f}, y) : \mathbf{f} \in E, y \in [0, J(\mathbf{f})]\}. \tag{79}$$

$G_{E,J}$ is compact in $\mathbb{R}^{n+1}$, hence there is a finite cover of open rectangles $\{R'_i\}$ satisfying $\mu(\cup_i R'_i) - \mu(G_{E,J}) < \frac{\epsilon}{8}$ on $\mathbb{R}^n$. Take their closures, and extend the sides of all rectangles indefinitely. This

results in a set of pairwise almost disjoint rectangles $\{R_i\}$. Taking only the rectangles $R = \{R_i : \mu(R_i \cap G_{E,J}) > 0\}$ results in a finite cover satisfying:

$$\sum_{i=1}^{|R|} \mu(R_i) - \mu(G_{E,J}) < \frac{\epsilon}{8}. \tag{80}$$

This implies:

$$\sum_{i=1}^{|R|} \mu(R_i) < \|J\|_1 + \frac{\epsilon}{8}, \tag{81}$$

and also,

$$\frac{\epsilon}{8} > \sum_{i=1}^{|R|} \int_{\mathbb{R}^n} \mathbb{1}_{R_i}(\mathbf{f}, J(\mathbf{f})) \, \mu(d\mathbf{f}) + \|J\|_1 \tag{82}$$

$$\geq \int_E |J(\mathbf{f}) - \sum_{i=1}^{|R|} \mathbb{1}_{R_i}(\mathbf{f}, J(\mathbf{f}))| \, \mu(d\mathbf{f}), \tag{83}$$

by the triangle inequality. For each $R_i = [a_{i1}, b_{i1}] \times \ldots [a_{in}, b_{in}] \times [\zeta_i, \zeta_i + y_i]$, let $X_i$ be its first $n$ components (i.e., the projection of $R_i$ onto $\mathbb{R}^n$). Then we have

$$\int_E |J(\mathbf{f}) - \sum_{i=1}^{|R|} y_i \mathbb{1}_{X_i}(\mathbf{f})| \, \mu(d\mathbf{f}) < \frac{\epsilon}{8}. \tag{84}$$

Let $Y(\mathbf{f}) \triangleq \sum_{i=1}^{|R|} y_i \mathbb{1}_{X_i}(\mathbf{f})$. By the triangle inequality,

$$\int_{\mathbb{R}^n} |J(\mathbf{f}) - \mathcal{K}(\mathbf{f})| \, \mu(d\mathbf{f}) \leq \|J - \mathbb{1}_E J\|_1 + \|\mathbb{1}_E J - Y\|_1 + \|\mathcal{K} - Y\|_1 \tag{85}$$

$$< \frac{\epsilon}{4} + \|\mathcal{K} - Y\|_1, \tag{86}$$

by (78) and (84). So it remains to construct $\mathcal{K}$ such that $\|\mathcal{K} - Y\|_1 < \frac{\epsilon}{4}$. Because $\mathbb{1}_{X_i}$ is discontinuous at the boundary of the rectangle $X_i$, it cannot be produced directly from a DI-Net (recall that all layers are continuous maps). However, we can approximate it arbitrarily well with a piece-wise linear function that rapidly ramps from 0 to 1 at the boundary.

For fixed rectangle $X_i$ and $\delta \in (0, 0.5)$, consider the inner rectangle $X_\delta \subset X_i$:

$$X_\delta = (a_1 + \delta(b_1 - a_1), b_1 - \delta(b_1 - a_1)) \times \cdots \times (a_n + \delta(b_n - a_n), b_n - \delta(b_n - a_n)), \tag{87}$$

where we omit subscript $j$ for clarity. Letting $b_i' = b_i - \delta(b_i - a_i)$, define the function:

$$T(\mathbf{f}) = \prod_{i=1}^{n} \frac{1}{\delta} \big[ \texttt{ReLU}(\delta - \texttt{ReLU}(\mathbf{f}_i - b_i')) - \texttt{ReLU}(\delta - \texttt{ReLU}(\mathbf{f}_i - a_i)) \big], \tag{88}$$

where $\texttt{ReLU}(x) = \max(x, 0)$. $T(\mathbf{f})$ is a piece-wise linear function that ramps from 0 at the boundary of $X_i$ to 1 within $X_\delta$, and vanishes outside $X_i$. Note that

$$\|\mathbb{1}_X - T\|_1 < \mu(X) - \mu(X_\delta) \tag{89}$$
$$= (1 - (1 - 2\delta)^n)\mu(X), \tag{90}$$

if $\mu$ is the Lebesgue measure. $\delta$ may need to be smaller under other measures, but this adjustment is independent of the input $\mathbf{f}$ so it can be specified *a priori*.

Recall that the function we want to approximate is $Y(\mathbf{f}) = \sum_{i=1}^{|R|} y_i \mathbb{1}_{X_i}(\mathbf{f})$. We can build NF-Net layers $\mathcal{K} : \mathbf{f} \mapsto \mathcal{K}(\mathbf{f}) = \sum_{i=1}^{|R|} y_i T_i(\mathbf{f})$, since this only involves linear combinations and ReLUs.

Then,

$$\|\mathcal{K} - Y\|_1 = \int_{\mathbb{R}^n} \sum_{i=1}^{|R|} y_i \left(T_i(\mathbf{f}) - \mathbb{1}_{X_i}(\mathbf{f})\right) d\mathbf{f} \tag{91}$$

$$= \sum_{i=1}^{|R|} y_i \|\mathbb{1}_{X_i} - T_i\|_1 \tag{92}$$

$$< (1 - (1 - 2\delta)^n) \sum_{i=1}^{|R|} y_i \mu(X_i) \tag{93}$$

$$= (1 - (1 - 2\delta)^n) \sum_{i=1}^{|R|} \mu(R_i) \tag{94}$$

$$< (1 - (1 - 2\delta)^n) \left(\|J\|_1 + \frac{\epsilon}{8}\right), \tag{95}$$

by (81). And so by choosing:

$$\delta = \frac{1}{2} \left(1 - \left(1 - \frac{\epsilon}{4} \left(\|J\|_1 + \frac{\epsilon}{8}\right)^{-1}\right)^{1/n}\right), \tag{96}$$

we have our desired bound $\|\mathcal{K} - Y\|_1 < \frac{\epsilon}{4}$ and thereby $\|J - \mathcal{K}\|_1 < \frac{\epsilon}{2}$.

*Case 2: $J$ is negative on some region of $\mathbb{R}^n$*

Letting $J^+(\mathbf{f}) = \max(0, J(\mathbf{f}))$ and $J^-(\mathbf{f}) = \max(0, -J(\mathbf{f}))$, define:

$$G_{E,J}^+ \triangleq \{(\mathbf{f}, y) : \mathbf{f} \in E, y \in [0, J^+(\mathbf{f})]\} \tag{97}$$

$$G_{E,J}^- \triangleq \{(\mathbf{f}, y) : \mathbf{f} \in E, y \in [0, J^-(\mathbf{f})]\}. \tag{98}$$

As in (80), construct covers of rectangles $R^+$ over $G_{E,J}^+$ and $R^-$ over $G_{E,J}^-$ each with bound $\frac{\epsilon}{16}$ and $\mathbb{R}^n$ projections $X^+$, $X^-$. Let:

$$Y^+(\mathbf{f}) = \sum_{i=1}^{|R^+|} y_i^+ \mathbb{1}_{X_i^+}(\mathbf{f}) \tag{99}$$

$$Y^-(\mathbf{f}) = \sum_{i=1}^{|R^-|} y_i^- \mathbb{1}_{X_i^-}(\mathbf{f}) \tag{100}$$

$$Y = Y^+ - Y^- \tag{101}$$

We can derive an equivalent expression to (84):

$$\frac{\epsilon}{8} > \int_E \left| J(\mathbf{f}) - \sum_{i=1}^{|R^+|} y_i^+ \mathbb{1}_{X_i^+}(\mathbf{f}) + \sum_{i=1}^{|R^-|} y_i^- \mathbb{1}_{X_i^-}(\mathbf{f}) \right| d\mathbf{f} \tag{102}$$

$$= \|\mathbb{1}_E J - Y\|_1. \tag{103}$$

Similarly to earlier, we use (78) and (103) to get:

$$\int_{\mathbb{R}^n} |J(\mathbf{f}) - \mathcal{K}(\mathbf{f})| \, d\mathbf{f} < \frac{\epsilon}{4} + \|\mathcal{K} - Y\|_1. \tag{104}$$

Choosing $T_i^+(\mathbf{f})$ and $T_i^-(\mathbf{f})$ the piece-wise linear functions associated with $X_i^+$ and $X_i^-$, and:

$$\mathcal{K}(\mathbf{f}) = \sum_{i=1}^{|R^+|} y_i^+ T_i^+(\mathbf{f}) - \sum_{i=1}^{|R^-|} y_i^- T_i^-(\mathbf{f}), \tag{105}$$

we have:

$$\|\mathcal{K} - Y\|_1 = \int_{\mathbb{R}^n} \left| \sum_{i=1}^{|R^+|} y_i^+ \left( T_i^+(\mathbf{f}) - \mathbb{1}_{X_i^+}(\mathbf{f}) \right) - \sum_{i=1}^{|R^-|} y_i^- \left( T_i^-(\mathbf{f}) - \mathbb{1}_{X_i^-}(\mathbf{f}) \right) \right| d\mathbf{f}, \qquad (106)$$

applying the triangle inequality,

$$\leq \sum_{i=1}^{|R^+|} y_i^+ \left\| \mathbb{1}_{X_i^+} - T_i^+ \right\|_1 + \sum_{i=1}^{|R^-|} y_i^- \left\| \mathbb{1}_{X_i^-} - T_i^- \right\|_1 \qquad (107)$$

$$< (1 - (1 - 2\delta^+)^n) \sum_{i=1}^{|R^+|} y_i^+ \mu(X_i^+) + (1 - (1 - 2\delta^-)^n) \sum_{i=1}^{|R^-|} y_i^- \mu(X_i^-) \qquad (108)$$

$$< (1 - (1 - 2\delta^+)^n) \left( \left\| J^+ \right\|_1 + \frac{\epsilon}{16} \right) + (1 - (1 - 2\delta^-)^n) \left( \left\| J^- \right\|_1 + \frac{\epsilon}{16} \right). \qquad (109)$$

By choosing:

$$\delta^+ = \frac{1}{2} \left( 1 - \left( 1 - \frac{\epsilon}{8} \left( \left\| J^+ \right\|_1 + \frac{\epsilon}{16} \right)^{-1} \right)^{1/n} \right) \qquad (110)$$

$$\delta^- = \frac{1}{2} \left( 1 - \left( 1 - \frac{\epsilon}{8} \left( \left\| J^- \right\|_1 + \frac{\epsilon}{16} \right)^{-1} \right)^{1/n} \right), \qquad (111)$$

and proceeding as before, we arrive at the same bounds $\|\mathcal{K} - Y\|_1 < \frac{\epsilon}{4}$ and $\|J - \mathcal{K}\|_1 < \frac{\epsilon}{2}$.

Putting it all together, Algorithm 1 implements the network logic for producing the function $\mathcal{K}$.

---

**Algorithm 1:** DI-Net approximation of $\mathbf{f} \mapsto J(\mathbf{f})$

---

*Setup*;
**Input:** target function $J$, $L_1$ tolerance $\epsilon/2$
Choose rectangles $R_i^+ = [a_{i1}^+, b_{i1}^+] \times \ldots [a_{in}^+, b_{in}^+] \times [\zeta_i^+, \zeta_i^+ + y_i^+]$ satisfying (80) and $R^-$
  similarly;
$\delta^+ \leftarrow \frac{1}{2} \left( 1 - (1 - \frac{\epsilon}{8} \left( \left\| J^+ \right\|_1 + \frac{\epsilon}{16} \right)^{-1})^{1/n} \right)$;
$\delta^- \leftarrow \frac{1}{2} \left( 1 - (1 - \frac{\epsilon}{8} \left( \left\| J^- \right\|_1 + \frac{\epsilon}{16} \right)^{-1})^{1/n} \right)$;

*Inference*;
**Input:** discretized input $\mathbf{f} = \{\mathbf{f}_k\}_{k=1}^n$
$x \leftarrow (0, 0, 1, 0, 0)$;
**for** rectangle $R_i^+ \in R^+$ **do**
 **for** dimension $k \in 1 : n$ **do**
  $x \leftarrow (\mathbf{f}_k - b_{ik}^+ + \delta(b_{ik}^+ - a_{ik}^+), \mathbf{f}_k - a_{ik}^+, x_3, x_4, x_5)$;
  $x \leftarrow \text{ReLU}(x)$;
  $x \leftarrow (\delta - x_1, \delta - x_2, x_3, x_4, x_5)$;
  $x \leftarrow \text{ReLU}(x)$;
  $x \leftarrow (0, 0, x_3(x_1 - x_2)/\delta, x_4, x_5)$;
 **end**
 $x \leftarrow (0, 0, 1, y_i^+ x_3 + x_4, x_5)$;
**end**
**for** rectangle $R_i^- \in R^-$ **do**
 **for** dimension $k \in 1 : n$ **do**
  $x \leftarrow (\mathbf{f}_k - b_{ik}^- + \delta(b_{ik}^- - a_{ik}^-), \mathbf{f}_k - a_{ik}^-, x_3, x_4, x_5)$;
  $\ldots$;
 **end**
 $x \leftarrow (0, 0, 1, x_4, y_i^- x_3 + x_5)$;
**end**
**Output:** $x_4 - x_5$

---

We can provide $x$ with access to $\mathbf{f}$ either through skip connections or by appending channels with the values $\{c + \mathbf{f}_k\}_{k=1}^n$ (which will be preserved under ReLU).

$\square$

**Theorem 3** (Maps between Single-Channel NFs). *For any Lipschitz continuous map $\mathcal{R} : \mathcal{F}_1 \to \mathcal{F}_1$, any $\epsilon > 0$, and any finite measure $\nu$ w.r.t. the measurable space $(\mathcal{F}_1, \mathscr{A}_{\epsilon, \mathcal{R}})$, there exists a DI-Net $\mathcal{T}$ that satisfies:*

$$\int_{\mathcal{F}_1} \|\mathcal{R}(f) - \mathcal{T}(f)\|_{L^1(\Omega)} \nu(df) < \epsilon. \tag{112}$$

*Proof.* If $\nu$ is not normalized, the discrepancy of our point set needs to be further divided by $\max\{\nu(\mathcal{F}_1), 1\}$. We assume for the remainder of this section that $\nu$ is normalized. Perform the construction of Lemma 5 $N$ times, each with a tolerance of $\epsilon/2NK$, where $K$ is the Lipschitz constant of $\mathcal{R}$. Choose a partition of unity $\{\psi_j\}_{j=1}^N$ for which $\psi_j(x) = \mathbb{1}\left[x_k = \arg\min_{x' \in X} d(x, x')\right]$, and output $N$ channels with the values $\{\mathcal{K}_j(\mathbf{f}) \psi_j(\cdot)\}_{j=1}^N$. By summing these channels we obtain a network $\tilde{\mathcal{K}}$ that fully specifies the desired behavior of $\tilde{\mathcal{R}} : \mathbb{R}^N \to \mathbb{R}^N$, with combined error:

$$\int_{\mathbb{R}^N} \left\| \tilde{\mathcal{R}}(\mathbf{f}) - \tilde{\mathcal{K}}(\mathbf{f}) \right\|_{\ell^1} \mu(d\mathbf{f}) < \frac{\epsilon}{2}. \tag{113}$$

Thus,

$$\int_{\mathcal{F}_1} \left| \frac{1}{|X|} \sum_{x' \in X} \tilde{\mathcal{R}} \circ \pi[f](x') - \tilde{\mathcal{K}} \circ \pi[f](x) \right| \nu(df) \leq \frac{\epsilon}{2}. \tag{114}$$

By (66) we have:

$$\int_{\mathcal{F}_1} \left| \int_\Omega \left| \pi^{-1} \circ \tilde{\mathcal{R}} \circ \pi[f](x) - \pi^{-1} \circ \tilde{\mathcal{K}} \circ \pi[f](x) \right| dx \right| \nu(df) \leq \frac{\epsilon}{2} + \frac{\epsilon}{6(M+2)} \tag{115}$$

By Lemma 3 we have:

$$\int_{\mathcal{F}_1} \int_\Omega \left| \mathcal{R}[f](x) - \pi^{-1} \circ \tilde{\mathcal{K}} \circ \pi[f](x) \right| dx\, \nu(df) \leq \frac{\epsilon}{2} + \frac{\epsilon}{6(M+2)} + \frac{\epsilon}{6} \tag{116}$$

And thus the network $\mathcal{T} = \pi^{-1} \circ \tilde{\mathcal{K}} \circ \pi$ gives us the desired bound:

$$\int_{\mathcal{F}_1} \|\mathcal{R}(f) - \mathcal{T}(f)\|_{L^1(\Omega)} \nu(df) < \epsilon. \tag{117}$$

$\square$

**Corollary 2** (Maps from NFs to vectors). *For any Lipschitz continuous map $\mathcal{R} : \mathcal{F}_1 \to \mathbb{R}^n$, any $\epsilon > 0$, and any finite measure $\nu$ w.r.t. the measurable space $(\mathcal{F}_1, \mathscr{A}_{\epsilon, \mathcal{R}})$, there exists a DI-Net $\mathcal{T}$ that satisfies:*

$$\int_{\mathcal{F}_1} \|\mathcal{R}(f) - \mathcal{T}(f)\|_{\ell_1(\mathbb{R}^n)} \nu(df) < \epsilon. \tag{118}$$

*Proof.* Let $M_0$ be the Lipschitz constant of $\mathcal{R}$ in the sense that $d(\mathcal{R}[f], \mathcal{R}[g])_{\ell^1} \leq M_0 d(f, g)_{L^1}$. Let $M = \max\{M_0, 1\}$. There exists $\tilde{\mathcal{R}} : \pi\mathcal{F}_1 \to \mathbb{R}^n$ such that $\left\| \tilde{\mathcal{R}} \circ \pi[f] - \mathcal{R}[f] \right\|_{\ell^1} \leq \epsilon/12$. As in Lemma 4, consider the extension of $\tilde{\mathcal{R}}$ to $\mathbb{R}^N \to \mathbb{R}^n$ in which each component of the output has the form:

$$\tilde{\mathcal{R}}_j(\mathbf{f}) = \begin{cases} \mathcal{R}[\pi^{-1}\mathbf{f}]_j & \text{if } \mathbf{f} \in \pi\mathcal{F}_1 \\ 0 & \text{otherwise.} \end{cases} \tag{119}$$

Then for similar reasoning, $\nu$ on $\mathcal{F}_1$ induces a measure $\mu$ on $\mathbb{R}^N$ that is finite and absolutely continuous w.r.t. the Lebesgue measure, and $\int_{\mathbb{R}^N} |\tilde{\mathcal{R}}_j(\mathbf{f})| \mu(d\mathbf{f}) < \infty$ for each $j$.

We construct our $\mathbb{R}^N \to \mathbb{R}$ approximation $n$ times with a tolerance of $\epsilon/2n$, such that:

$$\int_{\mathbb{R}^N} \left\| \tilde{\mathcal{R}}(\mathbf{f}) - \tilde{\mathcal{K}}(\mathbf{f}) \right\|_{\ell^1(\mathbb{R}^n)} \mu(d\mathbf{f}) < \frac{\epsilon}{2}. \tag{120}$$

Applying (65), we find that the network $\mathcal{T} = \tilde{\mathcal{K}} \circ \pi$ gives us the desired bound:

$$\int_{\mathcal{F}_1} \|\mathcal{R}(f) - \mathcal{T}(f)\|_{\ell_1(\mathbb{R}^n)} \nu(df) < \epsilon. \tag{121}$$

$\square$

**Corollary 3** (Maps from vectors to NFs). *For any Lipschitz continuous map $\mathcal{R} : \mathbb{R}^n \to \mathcal{F}_1$ and any $\epsilon > 0$, there exists a DI-Net $\mathcal{T}$ that satisfies:*

$$\int_{\mathbb{R}^n} \|\mathcal{R}(x) - \mathcal{T}(x)\|_{L^1(\Omega)} dx < \epsilon. \tag{122}$$

*Proof.* Define the map $\tilde{\mathcal{R}} : \mathbb{R}^n \to \pi\mathcal{F}_1 \subset \mathbb{R}^N$ by $\tilde{\mathcal{R}} = \pi \circ \mathcal{R}$. Since $\tilde{\mathcal{R}}$ is bounded and compactly supported, $\int_{\mathbb{R}^N} |\tilde{\mathcal{R}}_i(x)| dx < \infty$ for each $i$.

We construct a $\mathbb{R}^n \to \mathbb{R}$ approximation $N$ times each with a tolerance of $\epsilon/2NK$ with $K$ the Lipschitz constant, such that:

$$\int_{\mathbb{R}^n} \left\| \tilde{\mathcal{R}}(x) - \tilde{\mathcal{K}}(x) \right\|_{L^1(\Omega)} dx < \frac{\epsilon}{2}. \tag{123}$$

Applying (66), we find that the network $\mathcal{T} = \pi^{-1} \circ \tilde{\mathcal{K}}$ gives us the desired bound:

$$\int_{\mathbb{R}^n} \|\mathcal{R}(x) - \mathcal{T}(x)\|_{L^1(\Omega)} dx < \epsilon. \tag{124}$$

$\square$

Denote the space of multi-channel NFs as $\mathcal{F}_c = \{f : \Omega \to \mathbb{R}^c : \int_{\Omega} \|f\|_1 dx < \infty, \ f_i \in \mathcal{F}_1 \text{ for each } i\}$. Denote the norm on this space as:

$$\|f\|_{\mathcal{F}_c} = \int_{\Omega} \sum_{i=1}^c |f_i(x)| dx. \tag{125}$$

**Definition 5** (Concatenation). *Concatenation is a map from two NF channels $f_i, f_j \in \mathcal{F}_1$ to $[f_i, f_j] \in \mathcal{F}_2$. The concatenation of NFs can be defined inductively to yield $\mathcal{F}_n \times \mathcal{F}_m \to \mathcal{F}_{n+m}$ for any $n, m \in \mathbb{N}$.*

All maps $\mathcal{F}_n \times \mathcal{F}_m \to \mathcal{F}_c$ can be expressed as a concatenation followed by a map $\mathcal{F}_{n+m} \to \mathcal{F}_c$. A map $\mathbb{R}^n \to \mathcal{F}_m$ is also equivalent to $m$ maps $\mathbb{R}^n \to \mathcal{F}_1$ followed by concatenation. Thus, we need only characterize the maps that take one multi-channel NF as input.

Considering the maps $\mathcal{F}_n \to \mathcal{F}_m$, we choose a lower discrepancy point set $X$ on $\Omega$ such that the Koksma–Hlawka inequality yields a bound of $\epsilon/12mn(M+2)$. Let $\pi$ project each component of the input to $\pi\mathcal{F}_1$, and $\pi^{-1}$ inverts this projection under some choice function. We take $\mathscr{A}'$ to be the product $\sigma$-algebra generated from this $\pi$: $\mathscr{A}' = \{E_1 \times \cdots \times E_c : E_1, \ldots, E_c \in \mathscr{A}\}$ where $\mathscr{A}$ is the $\sigma$-algebra on $\mathcal{F}_1$ from Definition 4.

**Corollary 4** (Maps between multi-channel NFs). *For any Lipschitz continuous map $\mathcal{R} : \mathcal{F}_n \to \mathcal{F}_m$, any $\epsilon > 0$, and any finite measure $\nu$ w.r.t. the measurable space $(\mathcal{F}_n, \mathscr{A}'_{\epsilon,\mathcal{R}})$, there exists a DI-Net $\mathcal{T}$ that satisfies:*

$$\int_{\mathcal{F}_n} \|\mathcal{R}(f) - \mathcal{T}(f)\|_{\mathcal{F}_m} \nu(df) < \epsilon. \tag{126}$$

*Proof.* The proof is very similar to that of Theorem 3. Our network now requires $nN$ maps from $\mathbb{R}^{mN} \to \mathbb{R}$ each with error $\epsilon/2mnN$. Summing the errors across all input and output channels yields our desired bound. $\square$

The multi-channel analogue of Corollary 2 is clear, and we state it here for completeness:

**Corollary 5** (Maps from multi-channel NFs to vectors). *For any Lipschitz continuous map $\mathcal{R}$ :* $\mathcal{F}_n \to \mathbb{R}^m$, *any $\epsilon > 0$, and any finite measure $\nu$ w.r.t. the measurable space $(\mathcal{F}_n, \mathscr{A}'_{\epsilon, \mathcal{R}})$, there exists a DI-Net $\mathcal{T}$ that satisfies:*

$$\int_{\mathcal{F}_n} \|\mathcal{R}(f) - \mathcal{T}(f)\|_{\ell_1(\mathbb{R}^m)} \nu(df) < \epsilon. \tag{127}$$

## C  PIXEL-BASED DI-NET LAYERS

Here we present a variety of layers that show how to generalize pixel-based networks (convolutional neural networks and vision transformers) to DI-Net equivalents. Many of the following layers were not directly used in our experiments, and we leave an investigation of their properties for future work. We use $c_{\text{in}}$ to denote the number of channels of an input NF and $c_{\text{out}}$ to denote the number of channels of an output NF.

**Convolution layer**    The convolution layer aggregates information locally and across channels. It has $c_{\text{in}} c_{\text{out}}$ learned filters $K_{ij}$, which are defined on some support $S$ which may be a ball or orthotope. The layer also learns scalar biases $b_j$ for each output channel:

$$g_j = \sum_{i=1}^{c_{\text{in}}} K_{ij} * f_i + b_j, \tag{128}$$

with $*$ the continuous convolution as in (8).

To transfer weights from a discrete convolutional layer, $K$ can be parameterized as a rectangular B-spline surface that interpolates the weights (Fig. 2 left). To replicate the behavior of a discrete convolution layer with odd kernel size, $S$ is zero-centered. For even kernel size, we shift $S$ by half the dimensions of a pixel. We use a 2nd order B-spline for $3 \times 3$ filters and 3rd order for larger filters. We use deBoor's algorithm to evaluate the spline at intermediate points.

Strided convolution is implemented by simply truncating the output discretization to the desired factor as described in Section 4. Different padding behaviors from the discrete case are treated differently. Zero-padding is replicated by scaling $\mathcal{H}[f](x)$ by $\frac{|(S+x) \cap \Omega|}{S+x}$ where $S + x$ is the kernel support $S$ translated by $x$. For reflection padding, the value of the NF at points outside its domain are calculated by reflection. For no padding, the NF's domain is reduced accordingly, dropping all sample points that are no longer on the new domain.

**Linear combinations of channels**    Linear combinations of channels mimic the function of $1 \times 1$ convolutional layers in discrete networks. For learned scalar weights $W_{ij}$ and biases $b_j$:

$$g_j(x) = \sum_{i=1}^{c_{\text{in}}} W_{ij} f_i(x) + b_j, \tag{129}$$

for all $x \in \Omega$. These weights and biases can be straightforwardly copied from a $1 \times 1$ convolutional layer to obtain the same behavior. One can also adopt a normalized version, sometimes used in attention-based networks:

$$W_{ij} = \frac{w_{ij}}{\sum_{k=1}^{c_{\text{in}}} w_{kj}} \tag{130}$$

**Normalization**    All forms of layer normalization readily generalize to the continuous setting by estimating the statistics of each channel with numerical integration, then applying point-wise operations. These layers typically rescale each channel to have some mean $m_i$ and standard deviation $s_i$.

$$\mu_i = \int_{\Omega} f_i(x) dx \tag{131}$$

$$\sigma_i^2 = \int_{\Omega} f_i(x)^2 dx - \mu_i^2 \tag{132}$$

$$g_i(x) = \frac{f_i(x) - \mu_i}{\sigma_i + \epsilon} \times s_i + m_i, \tag{133}$$

where we assume $dx$ is normalized and $\epsilon > 0$ is a small constant. Just as in the discrete case, $\mu_i$ and $\sigma_i^2$ can be a moving average of the means and variances observed over the course of training different NFs, and they can also be averaged over a minibatch of NFs (batch normalization) or calculated per datapoint (instance normalization). Mean $m_i$ and standard deviation $s_i$ can be learned directly (batch normalization), conditioned on other data (adaptive instance normalization), or fixed at 0 and 1 respectively (instance normalization). These layers are not discretization invariant in the sense of Definition 1, since the output can be poorly behaved for small values of $\sigma_i$, but the convergence condition still holds, i.e. normalization is convergent under an equidistributed sequence of discretizations.

**Max pooling**    There are two natural generalizations of the max pooling layer to a collection of points: 1) assigning each point to the maximum of its k nearest neighbors, and 2) taking the maximum value within a fixed-size window around each point. However, both of these specifications change the output's behavior as the density of points increases. In the first case, nearest neighbors become closer together so pooling occurs over smaller regions where there is less total variation in the NF. In the second case, the empirical maximum increases monotonically as the NF is sampled more finely within each window. Because we may want to change the number of sampling points on the fly, both of these behaviors are detrimental.

If we consider the role of max pooling as a layer that shuttles gradients through a strong local activation, then it is sufficient to use a fixed-size window with some scaling factor that mitigates the impact of changing the number of sampling points. Consider the following simplistic model: assume each point in a given patch of an NF channel is an i.i.d. sample from $\mathcal{U}([-b, b])$. Then the maximum of $N$ samples $\{f_i(x_j)\}_{j=1}^N$ is on average $\frac{N-1}{N+1}b$. So we can achieve an "unbiased" max pooling layer by taking the maximum value observed in each window and scaling it by $\frac{N+1}{N-1}$ (if $N = 1$ or our empirical maximum is negative then we simply return the maximum), then (optionally) multiplying a constant to match the discrete layer.

To replicate the behavior of a discrete max pooling layer with even kernel size, we shift the window by half the dimensions of a pixel, just as in the case of convolution.

**Tokenization**    A tokenization layer chooses a finite set of non-overlapping regions $\omega_j \subset \Omega$ of equal measure such that $\cup_j \omega_j = \Omega$. We apply the indicator function of each set to each channel $f_i$. An embedding of each $f_i|_{\omega_j}$ into $\mathbb{R}^n$ can be obtained by taking its inner product with a polynomial function whose basis spans each $L^2(\omega_j)$. To replicate a pre-trained embedding matrix, we interpolate the weights with B-spline surfaces.

**Average pooling**    An average pooling layer performs a continuous convolution with a box filter, followed by downsampling. To reproduce a discrete average pooling with even kernels, the box filter is shifted, similarly to max pooling.

An adaptive average pooling layer can be replicated by tokenizing the NF and taking the mean of each token to produce a vector of the desired size.

**Attention layer**    There are various ways to replicate the functionality of an attention layer. Here we present an approach that preserves the domain. For some $d_k \in \mathbb{N}$ consider a self-attention layer with $c_{\text{in}} d_k$ parametric functions $q_{ij} \in L^2(\Omega)$, $c_{\text{in}} d_k$ parametric functions $k_{ij} \in L^2(\Omega)$, and a convolution with $d_k$ output channels, produce the output NF $g$ as:

$$Q_j = \langle q_{ij}, f_i \rangle \tag{134}$$

$$K_j = \langle k_{ij}, f_i \rangle \tag{135}$$

$$V[f] = \sum_{i=1}^{c_{\text{in}}} v_{ij} * f_i + b_j \tag{136}$$

$$g(x) = \texttt{softmax}\left(\frac{QK^T}{\sqrt{d_k}}\right) V[f](x) \tag{137}$$

A cross-attention layer generates queries from a second input NF. A multihead-attention layer generates several sets of $(Q, K, V)$ triplets and takes the softmax of each set separately.

**Data augmentation**    Most data augmentation techniques, including spatial transformations, point-wise functions and normalizations, translate naturally to NFs. Furthermore, spatial transformations are efficient and do not incur the usual cost of interpolating back to the grid. Thus DI-Nets might be suitable for a new set of data augmentation methods such as adding Gaussian noise to the discretization coordinates.

**Positional encoding**    Given their central role in neural fields, positional encodings (adding sinusoidal functions of the coordinates to each channel) would likely play an important role in helping pixel-based DI-Nets capture high-frequency information under a range of discretizations.

**Vector decoders ($\mathbb{R}^n \to \mathcal{F}_c$) and parametric functions in $\mathcal{F}_c$**    A vector can be expanded into an NF in several ways. We can create an NF that simply places input values at fixed coordinates and produces values at all other coordinates by interpolation. Alternatively, we can define a parametric function that spans $\mathcal{F}_c$ using the input vector as the parameters, for example by taking as input $n$ numbers and treating them as coefficients of the first $n$ elements of an orthonormal polynomial basis on $\Omega$. If $\Omega$ is a subset of $[a, b]^d$, one can use a separable basis defined by the product of rescaled 1D Legendre polynomials along each dimension. If $\Omega$ is a $d$-ball, we can use the Zernike polynomial basis. For a general coordinate system, a small MLP could be used where $\mathbb{R}^n$ can represent its parameters or a learned lower-dimensional modulation (Dupont et al., 2022) of its parameters. Beyond using such parametric functions as vector decoder layers, they also give rise to $n$-parameter layers that compute an inner product ("learned global pooling layer") or elementwise product ("dense modulation layer") of an input NF with the learned functions.

**Warp layer**    Layers that apply a self-homeomorphism $q$ on $\Omega$ (a bicontinuous map from $\Omega \to \Omega$) preserve discretization invariance since it simply modifies the upper bound of the invariance error in subsequent layers to use a discrepancy of $q(X)$ rather than a discrepancy of $X$.

# D    EXPERIMENTAL DETAILS

## D.1    ALGORITHMS

---
**Algorithm 2:** Classifier Training

---
**Input:** network $\mathcal{T}_\theta$, NF / label dataset $\mathcal{D}$, classifier loss $\mathcal{L}$, input discretization $X$
**for** step $s \in 1 : N_{\text{steps}}$ **do**
  NFs $f_i$, labels $y_i \leftarrow \text{minibatch}(\mathcal{D})$
  Label estimates $\hat{y}_i \leftarrow \mathcal{T}_\theta[f_i; X]$
  Update $\theta$ based on $\nabla_\theta \mathcal{L}(\hat{y}_i, y_i)$
**end**
**Output:** trained network $\mathcal{T}_\theta$

---

---
**Algorithm 3:** Dense Prediction Training

---
**Input:** network $\mathcal{T}_\theta$, dataset $\mathcal{D}$ with dense coordinate-label pairs, task-specific loss $\mathcal{L}$
**for** step $s \in 1 : N_{\text{steps}}$ **do**
  NFs $f_i$, point labels $(\boldsymbol{x}_{ij}, y_{ij}) \leftarrow \text{minibatch}(\mathcal{D})$
  Output NFs $g_i \leftarrow \mathcal{T}_\theta[f_i]$
  Point label estimates $\hat{y}_{ij} \leftarrow g_i(\boldsymbol{x}_{ij})$
  Update $\theta$ based on $\nabla_\theta \mathcal{L}(\hat{y}_{ij}, y_{ij})$
**end**
**Output:** trained network $\mathcal{T}_\theta$

---

## D.2    DETAILS ON IMAGENET CLASSIFICATION

We split ImageNet1k into 12 superclasses (dog, structure/construction, bird, clothing, wheeled vehicle, reptile, carnivore, insect, musical instrument, food, furniture, primate) based on the `big_12` dataset (Engstrom et al., 2019), which is in turn derived from the WordNet hierarchy.

We fit SIREN (Sitzmann et al., 2020b) to each image in ImageNet using 5 fully connected layers with 256 channels and sine non-linearities, trained for 2000 steps with an Adam optimizer at a learning rate of $10^{-4}$. It takes coordinates on $[-1, 1]^2$ and produces RGB values in $[-1, 1]^3$. We fit Gaussian Fourier feature (Tancik et al., 2020b) networks using 4 fully connected layers with 256 channels with ReLU activations. It takes coordinates on $[0, 1]^2$ and produces RGB values in $[0, 1]^3$.

The average pixel-wise error of SIREN is $3 \cdot 10^{-4} \pm 2 \cdot 10^{-4}$, compared to $1.6 \cdot 10^{-2} \pm 8 \cdot 10^{-3}$ for Gaussian Fourier feature networks. The difference in quality is visible at high resolution, but indistinguishable at low resolution (Fig. D.1). DI-Net-2 uses strided MLP convolutions, a global average pooling layer, then two fully connected layers. DI-Net-4 adds a residual block with two MLP convolutions after the strided convolutions. We train all models with an AdamW optimizer (Loshchilov & Hutter, 2017).

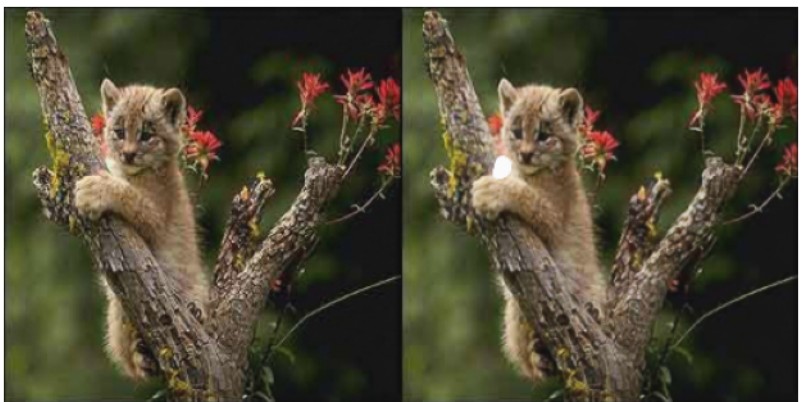

Figure D.1: Comparison of SIREN (left) and Gaussian Fourier feature network (right) representations of an image, rendered at $256 \times 256$ resolution. The Fourier feature network's representation is slightly blurrier (compare the tree bark), but this effect is not noticeable at lower resolutions.

The architecture of the MLP is 3 fully connected layers with 128 hidden units each and ReLU activation separated by batch normalization. It learns to map the SIREN parameters to the class label. We found that the model's loss curve becomes unstable after 3000 iterations so we reduce the number of iterations to 2000.

The non-uniform CNN applies the non-uniform Fourier transform (Muckley et al., 2020) followed by inverse Fast Fourier Transform to resample the input signal to the grid. It then feeds the result to a 2-layer CNN to perform classification.

During training, we augment with noise, horizontal flips, and coordinate perturbations.

### D.3    DETAILS ON CITYSCAPES SEGMENTATION

SIREN is trained on Cityscapes images for 2500 steps, using the same architecture and settings as ImageNet. Seven segmentation classes are used for training and evaluation, labeled in the dataset as 'flat' (e.g. road), 'construction' (e.g. building), 'object' (e.g. pole), 'nature', 'sky', 'human', and 'vehicle'.

DI-Net-3 uses two MLP convolutional layers at the same resolution followed by channel mixing (pointwise convolution). There are 16, 32 and 32 channels in the intermediate features. The support of the kernels in the MLP convolutional layers is $.025 \times .05$ and $.075 \times .15$ respectively, to account for the wide images in Cityscapes being remapped to $[-1, 1]^2$.

DI-Net-5 uses a strided MLP convolution to perform downsampling and nearest neighbor interpolation for upsampling. There are 16 channels in all intermediate features. There is a residual connection between the higher resolution layers.

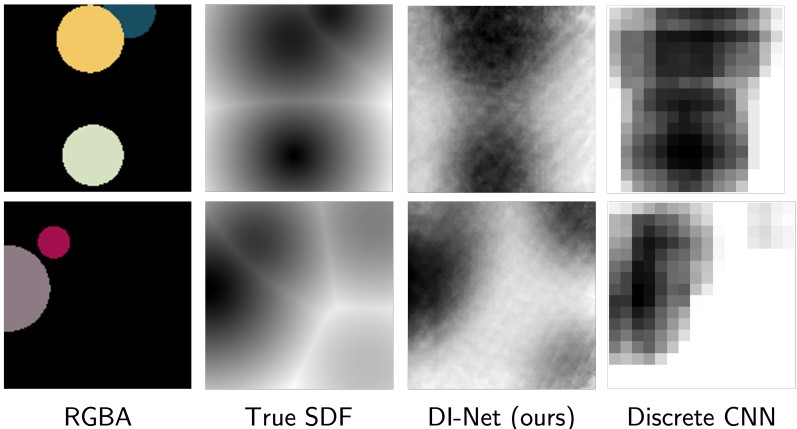

RGBA          True SDF          DI-Net (ours)          Discrete CNN

Figure D.2: 2D slices of two toy 3D scenes with signed distance functions predicted by DI-Net and a fully convolutional network.

The CNN baselines use 3x3 convolutions with the same number of layers and channels as DI-Net. All networks use ReLU activation and batch normalization.

The hypernetwork learns a map from the SIREN RGB to a SIREN with the same architecture that represents the segmentation. It predicts changes to the weights of all layers before the final fully connected layer, and predicts raw values for the weights of the final layer since it has 7 output channels for segmentation instead of 3 for RGB.

The non-uniform CNN applies the non-uniform Fourier transform followed by inverse Fast Fourier Transform, and feeds the result to the 3-layer FCN to perform segmentation.

## D.4    SIGNED DISTANCE FUNCTIONS

In our SDF prediction experiment, we construct toy scenes of 2-4 balls of random radii (range 0.2-0.5), centers, and colors scattered in 3D space ($\Omega = [-1, 1]^3$). For simplicity, we train each network directly on the closed form expressions for the RGBA fields and signed distance functions, rather than fitting neural fields first.

The FCN contains 3 convolutional layers of kernel lengths 3, 5 and 1 respectively. Accordingly, the convolutional DI-Net contained 2 convolutional layers followed by a linear combination layer. There are 8 channels in all intermediate features. We train each network for 1000 iterations with an AdamW optimizer with a batch size of 64 and a learning rate of 0.1 with an MSE loss on the SDF.

## E    ADDITIONAL ANALYSIS

### E.1    INITIALIZATION WITH DISCRETE NETWORKS

When a DI-Net is initialized with a large pre-trained convolutional neural network, its outputs are identical by construction. However, the behavior of the pre-trained CNN is not preserved when the DI-Net switches to other discretizations – even tiny perturbations from the regular grid are sufficient to change a classifier's predictions. Although the effect on the output of a single layer is much lower than the signal, small differences in each layer accumulate to exert a large influence on the final output. Figure E.2 illustrates this phenomenon for DI-Net initialized with a truncated EfficientNet. In addition, we find that once grid discretization is abandoned, large DI-Nets cannot easily be fine-tuned to restore the behavior of the discrete network used to initialize it. This suggests that not only does the discretization at training time not necessarily permit new discretizations at inference time, but also that the optimization landscape of maps on $L^2/I_X \to \mathbb{R}$ can vary significantly with $X$.

Table E.1: Pre-trained models fine-tuned on ImageNet NF classification.

| Model Type | Accuracy |
|---|---|
| EfficientNet (Tan & Le, 2019) | **66.4%** |
| DI-Net-EN | 48.1% |

Table E.2: Pre-trained models fine-tuned on Cityscapes segmentation.

| Model Type | Mean IoU | Pixel Accuracy |
|---|---|---|
| ConvNexT (Liu et al., 2022) | **0.429** | 68.1% |
| DI-Net-CN | 0.376 | **68.7%** |

In Tables E.1 and E.2, we illustrate that DI-Net initialized with a large pre-trained discrete network does not match the performance of the original model when fine-tuned with QMC sampling. We use a truncated version of EfficientNet (Tan & Le, 2019) for classification, and fine-tune for 200 samples per class. For segmentation we use a truncated version of ConvNexT-UPerNet (Liu et al., 2022), fine-tuning with 1000 samples.

We also find that the output of an NF-Net is less stable under changing sampling resolution with a grid pattern (Fig. E.1). While the output of a network with QMC sampling converges at high resolution, the grid sampling scheme has unstable outputs until very high resolution. Only the grids that overlap each other (resolutions in powers of two) produce similar activations.

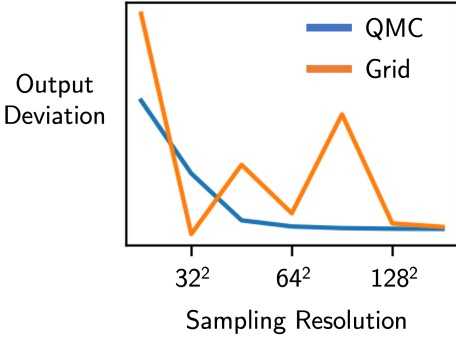

Figure E.1: Distance of the output of a DI-Net from its grid output at $32 \times 32$ resolution, when sampling at various resolutions. Its outputs deviate rapidly as its discretization shifts from a regular grid to a low discrepancy sequence

Figure E.2: An DI-Net's output diverges as sample points are gradually shifted from a grid layout to a low discrepancy sequence.

Our preliminary experience with DI-Nets highlights the need for improved sampling schemes and parameterizations that will allow large continuous-domain neural networks to learn effectively. Stable, scalable methods are needed to realize DI-Nets' full potential for continuous data analysis.

### E.2 COMPUTATIONAL COMPLEXITY

The DI-Net's complexity is similar to that of a discrete model with an equivalent architecture. In general time and memory both scale linearly with the number of sample points (regardless of the dimensionality of $\Omega$), as well as with network depth and width.

Implemented naively, the computational cost of the continuous convolution is quadratic in the number of sample points, as it must calculate weights separately for each neighboring pair of points. We can reduce this to a linear cost by specifying a $N_{\text{bin}}$ Voronoi partition of the kernel support $B$, then using the value of the kernel at each seed point for all points in its cell. Thus the kernel need only be evaluated $N_{\text{bin}}$ times regardless of the number of sample points. Additionally $N_{\text{bin}}$ can be modified during training and inference.

DI-Net-4 (our ImageNet classifier) performs a forward pass on a batch of 48 images in $96 \pm 4$ms on a single NVIDIA RTX 2080 Ti GPU.

## F  FUTURE DIRECTIONS

**Scaling convolutional DI-Nets**    Our initial experiments suggest that convolutional DI-Nets do not scale well in depth. We suspect that within a CNN-like architecture, the gradients of discrete convolutional layers with respect to kernel parameters have much smoother optimization landscapes over large networks relative to continuous convolutional layers parameterized with MLPs or coefficients of a polynomial basis. It is then no surprise that implementations of neural networks with continuous convolutions do not simply substitute the convolutional layers in a standard CNN architecture, but also make use of a variety of additional techniques (Qi et al., 2017; Wang et al., 2021; Boulch, 2019) which would likely be helpful for scaling convolutional DI-Nets.

**Parameterization of output NFs**    In this work we assume that a DI-Net that produces an NF specifies the output discretization *a priori*, but some applications may need the output to be sampled several times at different discretizations. It is inefficient to re-evaluate the entire network in such cases, and we propose two solutions for future work. One method can store the last few layers of the network alongside the input activation, and adapt the discretizations as needed in these last few layers only. A second approach can treat the discretized outputs of DI-Net as parameters of the output NF in the manner of Vora et al. (2021), which would maintain interoperability of the entire framework.

**Extending DI-Net to high discrepancy sequences**    In many applications, there are large regions of the domain that are less informative for the task of interest. For example, most of the information in 3D scenes is concentrated at object surfaces, so DI-Nets should not need to process a NeRF by densely sampling all 5 dimensions. Moreover, ground truth labels for dense prediction tasks may only be available along a high discrepancy discretization. Such a discretization can be handled by quadrature, but more work is required to design efficient quadrature methods within a neural network. Additional techniques such as learned coordinate transformations or learned discretizations may also be helpful for extending our model to extreme discretizations or highly non-uniform measures.

**Error propagation**    When an NF does not faithfully represent the underlying data, it is important to characterize the influence on DI-Net's output. In the worst case, these deviations are adversarial examples, and robustness techniques for discrete networks can also be applied to DI-Net. But what can we say about typical deviations of NFs? Future work should analyze patterns in the mistakes that different types of NFs make, and how to make DI-Nets robust to these.

