# OpenReview forum: "Discretization Invariant Learning on Neural Fields"
_ICLR.cc/2023/Conference — Submitted to ICLR 2023_

### Official Review · Reviewer_XJni · 2022-10-17

**Confidence:** 4
**Correctness:** 3
**Technical Novelty And Significance:** 3
**Empirical Novelty And Significance:** 3
**Recommendation:** 6

**Clarity, Quality, Novelty And Reproducibility:**

**Clarity**

The paper is well written and the figures are nice. This is likely a personal preference, but the paper often feels unnecessarily formal, when a more intuitive approach might have been more effective.

**Quality**

The paper is well presented, experiments are quite thorough (although small scale) and the paper openly acknowledges limitations. As such, I believe this is quite a high quality paper.

**Novelty**

The proposed method is novel in the sense that it provides a new way to handle deep learning on neural fields. However, the resulting models are, in practice, quite similar to existing models for continuous convolutions.

**Reproducibility**

The details and explanations provided in the paper seem to be sufficient for reproducing the results in the paper. The appendix is detailed.

**Details Of Ethics Concerns:**

No ethical concerns.

**Strength And Weaknesses:**

**Strengths**

- As neural fields are becoming increasingly popular, the paper tackles an interesting and timely problem, namely that of performing various machine learning tasks on neural fields.
- The proposed model is invariant to the parameterization of the neural field, which is a desirable property. The experiments training on SIRENs and testing on Fourier feature networks highlight this nicely.
- The paper is well written and polished and the figures are nice.
- The authors provide a thorough theoretical analysis of the proposed model.
- The appendix is thorough and well done, containing lots of details and exciting ideas for future work.

**Weaknesses**

- While the authors claim their model is discretization invariant, this statement is contradicted by experimental results. Of course, the model is discretization invariant in the sense of Definition 2, but the both the variation of f and discrepancy of X can be large, which means that the model is not discretization invariant in the practical sense of the term. This is demonstrated in numerous experiments throughout the paper, e.g. Table 2 and Table 4. I think it would therefore be good to temper the claims about discretization invariance or at least highlight early on that the theoretical definition of discretization invariance does not necessarily imply discretization invariance in practice.
- In general, it feels like the theory is so "general" that it doesn't really provide any particularly useful guidance on how to actually instantiate a model. Indeed, the resulting models are very similar to various continuous convolution models.
- The experimental results are quite weak (this is also acknowledged by the authors). The authors only perform small scale experiments and compare their results to fairly weak baselines (e.g. a 2 layer CNN). While it is okay to underperform CNNs designed for image classification, it would be interesting to include comparisons to other neural field based approaches (or explain why they are not relevant).
- One of the advantages of using neural fields is that they typically provide a very compact representation of data (which was the motivation for the original 3D works on neural fields). However, using DI-Nets, we are back to sampling the original "data space" which, for 3D experiments for example, still would require a very large number of samples and so defeats the purpose of using the compact neural field representation in the first place. In contrast, other methods that act directly on some compact parameterization of a neural field are likely to scale better. The scaling of DI-Nets is then similar to just acting directly on the discrete signal (as mentioned in the appendix, "DI-Net's complexity is similar to that of a discrete model with an equivalent architecture"). I believe it would be worth discussing this when comparing to related work.
- This might be a style preference, but the language/formulation sometimes seems unnecessarily formal, and a more intuitive presentation might have worked better.

**Typos**

- The figure reference on page 26 in the appendix is wrong (fig:layers left)


**Summary Of The Paper:**

This paper introduces DI-Nets, a formulation of neural networks that is invariant to discretization of the input space. More specifically, the authors consider building models that take neural fields as input, and unlike other recent work, do not depend on the specific parameterization of that neural field, which is desirable. The authors prove various theoretical properties of their model, including discretization invariance for the resulting networks and their gradients as well as a universal approximation theorem. The authors test an instantiation of their model in various toy experiments.


**Summary Of The Review:**

Overall, I think this is a fairly good paper, which will be of interest to the ICLR community. The paper is well written, the method is sound and the experiments are fairly interesting. However, there are some issues around discretization invariance as well as empirical performance in practice. I therefore recommend a weak accept.

---

> ### Author Response · Authors · 2022-11-19
> **Reply to Reviewer XJni**
>
> We thank the reviewer for their very thorough and thoughtful feedback. We are pleased with their comments that “the paper is well written, the method is sound and the experiments are fairly interesting”, and that we “tackle an interesting and timely problem” with an approach that “will be of interest to the ICLR community”.
>
> > “It would be good to temper the claims about discretization invariance or at least highlight early on that the theoretical definition of discretization invariance does not necessarily imply discretization invariance in practice”
>
> We agree that discretization invariance is achieved in only a partial sense, and we believe that highlighting where these gaps can occur is perhaps our most interesting empirical result. We revise our contributions in the introduction to add this nuance:
> *“We probe the limits of discretization invariance in practice, finding that DI-Net has some ability to generalize to new discretizations at test time, modulated by the task and the type of discretization(s) it was trained on.”*
>
> > “In general, it feels like the theory is so "general" that it doesn't really provide any particularly useful guidance on how to actually instantiate a model. Indeed, the resulting models are very similar to various continuous convolution models.”
>
> We have added a paragraph to the end of our universal approximation theorem linking our theory to guidance on DI-Net design, including alternatives to convolutional DI-Nets and how to choose between them. Our results suggest that the most obvious instantiation of DI-Net – taking a discrete model and extending it to the continuous domain – is sufficient to produce a working class of models with interesting properties. Since continuous convolutions are the discretization invariant extension of CNNs, our models will naturally have similarities with existing works that aim to be discretization-invariant CNNs. We hope that future work will also help reveal more concrete strategies for designing efficient and performant DI-Nets.
>
> > “It would be interesting to include comparisons to other neural field based approaches (or explain why they are not relevant)”
>
> We have added comparisons to other approaches for learning on neural fields.
> * On classification, we added an MLP that predicts class labels directly from SIREN parameters, and added non-uniform convolutions [1] as a representative of other approaches to processing non-grid data. We find that both approaches perform significantly worse than our method and CNN baseline.
> * On segmentation, we added a hypernetwork that maps the parameters of the SIREN image to the parameters of a new SIREN that represents its segmentation, as well as the non-uniform convolution approach in [1]. We find that these methods underperform on this task.
>
> These previous approaches also lack key benefits of our approach: the MLP/hypernetwork approach is not interoperable between different types of neural fields and cannot be applied to fields whose parameters extend beyond a neural network; the non-grid convolution [1] cannot preserve high-frequency information and only produces outputs along the grid unless other techniques are incorporated.
>
> [1] Jiang et al. Convolutional Neural Networks on non-uniform geometrical signals using Euclidean spectral transformation. ICLR 2019.
>
> > “Using DI-Nets, we are back to sampling the original "data space" which [...] still would require a very large number of samples”
>
> Great point. Our initial experiments focus on discretizations which are low discrepancy with respect to the uniform measure, which is indeed inefficient for sparse 3D scenes. The key for overcoming this limitation is to extend our framework to non-uniform (learned) measures, as we mention in Appendix F. For example, one approach could use an adaptive sampling scheme that targets more informative areas of the domain. We believe such extensions, although crucial for scaling such models to large 3D scenes, are best left for a more empirically-focused paper.
>
> > “The language/formulation sometimes seems unnecessarily formal” / “might benefit from a more intuitive approach”
>
> We agree that our presentation could be improved with more intuition. We have significantly restructured section 3 to focus on building intuition for our method and theoretical results.
>
> > “The figure reference on page 26 in the appendix is wrong”
>
> Thank you for the correction!

---

> > ### Comment · Reviewer_XJni · 2022-11-20
> > **Thank you for your response**
> >
> > Thank you for your thorough and detailed response! I appreciate the changes related to tempering the claims around discretization invariance as well as the rewrite of the paper focusing on a more intuitive approach. I also appreciate the inclusion of additional baselines. However, I also believe these baselines (e.g. directly predicting labels from MLP weights) are quite weak. For neural fields, I believe it would be more interesting to compare to e.g. the approach proposed in the "from data to functa" paper. For continuous convolutions, I believe it would be more interesting to compare to established models like PointConv which are widely used.
> >
> > Based on the updates to the paper as well as the other reviews I still believe this is a valuable paper. However, there are still some limitations (particularly empirical results - discretization invariance is not achieved in practice and baselines are quite weak) which means I maintain my score of weak accept.

---

> > > ### Author Response · Authors · 2022-11-20
> > > **Re: Thank you for your response**
> > >
> > > Thank you for your thoughtful feedback! We agree the baselines you raised are interesting, and would be important for an empirically focused paper that proposes a particular design for convolutional DI-Nets. We comment on why these baselines may not be as helpful in the context of our current paper:
> > >
> > > Implementing PointConv in place of our MLP convolution would likely yield a small boost in performance, although the main difference between PointConv and our parameterization is that they reweight different parts of the conv kernel based on the density of sample points. Such a design would only see benefits with high discrepancy sequences (which is useful with point clouds, but not in our setting with QMC). Indeed, we discovered a few other optimizations that would lead to small improvements in the performance of DI-Net (e.g. adding positional encodings), but we omitted them because we believe the best way to understand DI-Nets is to characterize how they behave in comparison to *architecturally equivalent* CNNs.
> > >
> > > The "data to functa" method is a good choice when one must build the neural field dataset from scratch. But we believe it is overly restrictive for our setup, where we would like to disentangle construction of the dataset from learning of the task (which is of course the usual case in machine learning). Hypernetworks and MLPs are already restrictive in the sense that they do not work on many types of neural fields (those with voxels, hash tables, octrees, etc.), and can only work on the specific neural field parameterization that it is trained on. "Data to functa" is *further* restricted to neural fields that are parameterized with "modulations" (so we would have needed to refit our ImageNet and Cityscapes datasets to different "modulated SIRENs"). It then fits simple MLPs to the modulations, and hence is almost entirely driven by the quality of the modulations (i.e. the quality of the constructed dataset) rather than the quality of the task-specific component. Thus, the modulation-based approach can be viewed as tackling an entirely different problem from our approach.

---

> > > > ### Comment · Reviewer_XJni · 2022-11-28
> > > > **Re: Re: Thank you for your response**
> > > >
> > > > Thank you for your response - these are all good points. In particular, I think the point about DI-Nets being applicable to many more types of neural fields (including ones with other data structures such as hash tables and octrees) is actually quite a significant strength of the proposed approach and it would be worthwhile highlighting this in the paper. Most papers I've read on processing neural fields (including concurrent ones submitted to this conference) are restricted to MLP architectures only and DI-Nets are therefore more flexible in that sense.

---

> > > > > ### Author Response · Authors · 2022-11-29
> > > > > **Thank you**
> > > > >
> > > > > Thank you for the kind comment! We agree that the wider applicability of DI-Net is a unique strength, and we will highlight this benefit more prominently in future versions of the paper.

---

### Official Review · Reviewer_NWbG · 2022-10-24

**Confidence:** 4
**Correctness:** 3
**Technical Novelty And Significance:** 2
**Empirical Novelty And Significance:** 2
**Recommendation:** 3

**Clarity, Quality, Novelty And Reproducibility:**

The paper is well structured, but I think it would be more helpful to add more details about
network implementation.
I think the DI-Net might not be stable, at least numerically. As the authors point out in
section 5.1, CNN outperforms DI-Net in low-resolution. And I think the superiority of DI-Net in
higher-resolution is not obvious enough to convince me. Table 3 also shows that DI-Net can not
defeat FCN under certain condition.
The authors did not provide their code, instead they offered some configurations of experiments,
but not all. They offered information about depth, learning rate, dataset, learning rate, optimizer
and discretization level. But details of the network implementation were not mentioned. One
might encounter difficulties when reproducing the results.

**Strength And Weaknesses:**

Strength
This paper gives a general idea of discretization invariant learning. It connects its key idea
DI-Nets with existing popular architectures, such as CNN and neural operators. It offers a general
perspective of these models.

Weakness
The pseudo code of algorithm 1 and 2 in page 7 is negligible as they are the standard procedure
of general learning process.
Neural operators, as the authors mentioned in the paper, is an important category of discretiza-
tion invariant network. But the numerical experiments focus mainly on the computer vision, e.g.
classification and segmentation. I think it would be better if there were an experiment on neural
operators. Personally, I think such experiments are an important aspect of discretization invariant
learning.
In the numerical experiments part, the authors mainly compare different methods of dis-
cretizaion, such as MC, QMC, grid and shrunk. But there were few attention on the discretization
level. There was only a partial comparison in Table 1 on 32*32 and 64*64. I think it would be
better if there were comparison between more sampling levels (of the same sampling method).

**Summary Of The Paper:**

This paper introduces a general framework of discretization invariant network, DI-Net. The
authors give a theoretical upper bound w.r.t. variation and discrepancy in the finite samples case.
They also reveal that DI-Net is a generalization of existing neural fields networks such as CNN.
They demonstrate the performance of DI-Net against CNN and FCN with different discretization
strategies.

**Summary Of The Review:**

I think this paper offers a general idea of discretization invariant learning on neural fields and
supplies plenty of theoretical results. However, I think the detail of the DI-Net is not abundant
enough for readers. Also, personally the numerical results are not good enough to convince me the
superiority of DI-Net.

---

> ### Author Response · Authors · 2022-11-19
> **Reply to Reviewer NWbG**
>
> We thank the reviewer for their feedback. We are pleased that the reviewer found our paper "well structured" with “plenty of theoretical results”. However we respectfully disagree with the reviewer’s assessment that our contributions are only marginally significant or novel.
>
> To clarify a potential miscommunication, the objective of this paper is **not** to improve on neural operators as implied by the review, but to propose an entirely new principled framework for learning on neural fields. To this end, we present a number of broad theoretical insights including sufficient conditions for discretization invariant layers and the quasi-Monte Carlo method as a natural choice for discretization invariance, among several others. We also observe and characterize an **entirely new phenomenon**: that choice of discretization can influence learning and generalization behavior differently under different tasks. We therefore believe that our framework and insights are interesting, novel and relevant to the broader ICLR community, as mentioned by other reviewers.
>
> We also address specific concerns as follows:
>
> > “One might encounter difficulties when reproducing the results.”
>
> We provide an anonymized version of our code base in the supplementary materials, with instructions for replicating our experiments. If our paper is accepted, we will add a link to the code repository. Additionally, we now include more experimental details in the appendix for completeness.
>
> > "The pseudo code of algorithm 1 and 2 in page 7 is negligible as they are the standard procedure of general learning process"
>
> We moved the pseudocode to the Appendix. We believe the pseudocode can still aid readers in understanding our method because these similarities ground it in a familiar scheme. The key distinction of our method from general learning processes is that our input discretization can always be chosen, and in dense prediction tasks, the output discretization is determined by the discretization of ground truth labels at training time, and it can be specified by the user at test time.
>
> > "The numerical experiments focus mainly on the computer vision, e.g. classification and segmentation. I think it would be better if there were an experiment on neural operators"
>
> We focus on computer vision tasks as vision and graphics are the areas where neural fields have the greatest impact and potential. The problem of learning maps between neural fields is the key motivation for this framework, as there is not always a natural way to discretize the field. We note that discretization invariance was not a goal for neural operators but rather an emergent property of their design (in fact they are a subfamily of DI-Nets). Most neural operators are also optimized to solve partial differential equations, and we do not aim to compete with them on that domain.
>
> > “But there were few attention on the discretization level. There was only a partial comparison in Table 1 on 3232 and 6464. I think it would be better if there were comparison between more sampling levels (of the same sampling method)”
>
> We agree such comparison is also interesting. We add a comparison between more sampling levels, and report the results in Fig. 3. We find that the generalization benefit of our method is consistent across even higher resolutions, and becomes much more pronounced at lower out-of-distribution resolutions, where the standard CNNs drop severely in performance whereas our methods are not degraded.
>
> > “Personally the numerical results are not good enough to convince me the superiority of DI-Net.”
>
> As stated in the opening of the experiments, we do **not** aim to compete with CNNs on computer vision benchmarks or to establish the superiority of DI-Net over existing work. Instead, we aim to develop a novel theoretically-sound method for learning on neural fields and to illustrate its unique characteristics and examine its behaviors on various discretizations, tasks, and datasets. DI-Net is a novel framework with a unique combination of traits: discretization invariance, interoperability between neural fields, the ability to choose the input and output discretizations independently, and outputs that are themselves neural fields. We further emphasize that we did not pursue an optimized design for DI-Net, so that we could instead compare it directly to CNNs with the same architecture to remove potential experimental confounders. Lastly, we propose a number of empirical optimizations in Appendix F, but we believe adding such techniques to this paper would detract from our focus on discretizations.

---

> ### Author Response · Authors · 2022-12-06
> **Follow-up to Reply to Reviewer NWbG**
>
> Dear Reviewer NWbG,
>
> We would like to gently remind you that the discussion period is ending soon. We would greatly appreciate it if you could please review our response and revisions, which we believe address the concerns raised in your initial review. We hope that we can address any remaining concerns and questions you may have to improve the quality of our paper.
>
> We understand that your time is valuable, and we appreciate your efforts in reviewing our paper. We would be grateful if you could provide your thoughts on our revised submission and potentially reconsider your rating if your concerns are sufficiently addressed. Thank you for your time and consideration.

---

### Official Review · Reviewer_6WWK · 2022-10-24

**Confidence:** 3
**Correctness:** 3
**Technical Novelty And Significance:** 3
**Empirical Novelty And Significance:** Not applicable
**Recommendation:** 6

**Clarity, Quality, Novelty And Reproducibility:**

The paper is novel and the quality of the work seems high
My main concerns are clarity and reproducibility

**Strength And Weaknesses:**

Strengths:
- The paper addresses an important problem, the dependency of standard learning approaches on discretization.
- The work in novel and very interesting

Weaknesses:
- The paper is not well written.
(1) Lots of space wasted on trivial points, like prop.1 that simply says that the function is well defined. Or the list of operations later that doesn't really helps understand the work proposed. Also Prop. 2 is trivial from definition (eq. 1).
(2) The main technical part of the paper is glossed over, 3 lines in bullet point 3 in page 6. Everything up to that point was simply replacing an integral with an average.
(3) The whole mathematical machinery feels redundant or under-utilized. How is it different then convolutions on a non-uniform grid?
- The experimental part should compare to previous works such as Jiang et al "CONVOLUTIONAL NEURAL NETWORKS ON NONUNIFORM GEOMETRICAL SIGNALS USING EUCLIDEAN SPECTRAL TRANSFORMATION"

**Summary Of The Paper:**

The paper proposes a novel approach for learning over neural fields that is discretization invariant. The authors propose a mathematical framework and design a network following this definition. They show experimental results on several vision benchmarks.

**Summary Of The Review:**

The paper proposes an interesting and novel approach to an important problem, but I do not think it is clear enough and I have very low confidence in my ability to reproduce the results from the papers description.

---

> ### Author Response · Authors · 2022-11-19
> **Reply to Reviewer 6WWK (1/2)**
>
> We thank the reviewer for their helpful suggestions and insights. We are pleased that they found our approach to be an interesting and novel approach to an important problem. Based on their feedback, we made numerous revisions to improve clarity and reproducibility and have added their suggested baseline. We also respond to the reviewer’s comments below:
>
> > Space wasted: "Prop.1 simply says that the function is well defined".
>
> We clarify that the key message of Prop. 1 was that our method is interoperable between different types of neural fields. We have removed Prop. 1 and now state that message directly.
>
> > “The list of operations later doesn't really helps understand the work proposed”.
>
> We agree that this could have been clearer. The list of operations originally presented the types of layers allowed in DI-Nets before focusing on one of the items in the list (discretization invariant layers). We now first describe DI layers and then define DI-Nets along a much more concise version of this list.
>
> > “Prop. 2 is trivial from definition (eq. 1).”
>
> Prop. 2 had two key messages: that equidistributed sequences are a natural generator of discretizations, and that DI-Net layers converge in the forward direction. We have removed Prop. 2 and now present these messages directly for conciseness.
>
> > "The main technical part of the paper is glossed over, 3 lines in bullet point 3 in page 6"
>
> We are glad the reviewer is interested in the proof of the universal approximation theorem for our method. We add more detail to the sketch in the main text and discuss why the proof is interesting.
>
> > “Everything up to that point was simply replacing an integral with an average”.
>
> Up to that point we presented several key insights and steps of reasoning to develop a discretization-invariant framework for neural fields, beyond simply replacing integrals with sample means. We hope that our restructuring of section 3 (summarized below) makes these insights/steps clearer:
> - We define variation, and describe neural fields as parametric integrable functions of bounded variation
> - We define discretization invariance, describe discrepancy, and state the Koksma–Hlawka inequality (“replacing an integral with an average”)
> - We propose specific functional forms of discretization invariant layers and describe necessary conditions (differentiability, etc.). We remark on interoperability of our framework, and implementing loss functions for DI-Nets.
> - We define DI-Nets, describe how different layer types fit together, note that discretization invariance favors quasi-Monte Carlo integration, and discuss alternative numerical integration techniques.
> - We describe convergence of DI-Net under equidistributed sequences, discuss the Gateaux derivative, and prove that network gradients also converge under such sequences to the Gateaux derivative with respect to a particular sequence of bump functions.
> Replacing an integral with its sample mean thus only accounts for a small part of our conceptual framework and theoretical results.
>
> > “The whole mathematical machinery feels redundant or under-utilized.”
>
> We have removed references to non-essential mathematical concepts such as ideal (quotient ring), weak derivative and Sobolev space. Other concepts such as variation, discrepancy, and Gateaux differentiability are described in more detail, and we justify why they are needed to obtain our theoretical results.
>
> In case the reviewer (also) means the theory is under-utilized in relation to our experiments, we believe this reflects the fact that DI-Nets define a very broad class of neural networks. We must focus on a particular instantiation of DI-Nets (in our case, small convolutional networks) in order to characterize its learning and generalization behaviors under various discretizations.
> We hope this paper will pave the way for future work (by us and others) that continues to utilize the fundamental properties of DI-Nets we have shown here and to specialize the analysis to narrow classes of DI-Nets relevant to different applications of interest.
>
> [continued...]

---

> > ### Author Response · Authors · 2022-11-19
> > **Reply to Reviewer 6WWK (2/2)**
> >
> >
> > > "How is it different then convolutions on a non-uniform grid?"
> >
> > Great question - there are several important distinctions. Previous works on continuous or non-uniform convolutions assumed that the discretization is always specified at both training and test time. Here we analyze the behavior of convolutional DI-Nets under various discretizations, building on top of our theoretical results (which also extend to non-convolutional settings). For dense prediction tasks, our framework can also evaluate the output anywhere on its domain, which is not typically supported by other approaches.
> >
> > Crucially, our framework *gives rise to continuous convolutions as the discretization invariant extension of CNNs*, so it will naturally have similarities with existing works that **specifically** aim to be discretization-invariant CNNs. We further clarify that the discretization-invariant CNN that arises from our method is closer to Wang et al.’s “Deep Parametric Continuous Convolutional Neural Networks” than to Jiang et al. Beyond convolutions, we offer a framework for building entire networks that operate on neural fields and that are invariant to discretization.
> >
> > Lastly, not all non-uniform convolutional networks are discretization invariant. For example, Jiang et al. still pass the signal through a regular CNN after converting spectral information back to the physical domain, hence there is no sufficiently fine discretization of the input that will preserve signals higher than a certain frequency.
> >
> > We have updated the paper to clarify these connections. Whereas our original submission described continuous convolutions as part of “Discretization invariant networks” under Related Work, we expanded this discussion into a separate paragraph.
> >
> > > “The experimental part should compare to previous works such as Jiang et al”
> >
> > We have added a comparison to Jiang et al.’s method for classification and segmentation, and find that it significantly underperforms our method and standard CNN baseline on both tasks. We also note that their method still relies on a standard discrete CNN / U-Net, so without using additional techniques it only produces predictions along the grid and does not preserve high frequency information. As noted in our general response, we also add baselines for learning on neural field parameters directly, which also underperforms.
> >
> > > “I have very low confidence in my ability to reproduce the results from the papers description.”
> >
> > We attach an anonymized version of our code base in the supplementary materials, with instructions for replicating our experiments. If our paper is accepted, we will add a link to the code repository. Further, we now include more experimental details in the appendix for completeness.

---

> > > ### Comment · Reviewer_6WWK · 2022-11-19
> > > **Missing revision**
> > >
> > > I don't see any revision nor does one appear in the revision history in November. I also can't see any submitted code.

---

> > > > ### Author Response · Authors · 2022-11-19
> > > > **Revision is up now**
> > > >
> > > > Sorry for the delay and confusion! The revisions and submitted code are now uploaded.

---

> > > > > ### Comment · Reviewer_6WWK · 2022-11-26
> > > > > **Thanks**
> > > > >
> > > > > The writing has improved, and the additional baselines I asked for were added so I raised my score. I would comment that the details on the additional experiments are lacking, for instance, why is everything trained with a 1e-3 learning rate? You should find the best LR on a validation set for each approach.

---

> > > > > > ### Author Response · Authors · 2022-11-29
> > > > > > **Thanks, updated baselines**
> > > > > >
> > > > > > Thank you for your support and good suggestion! We have rerun the additional baselines with hyperparameter tuning and were able to improve their performance (although they still do not reach the same level of performance as DI-Net and the discrete CNNs). In future versions of the paper we will report these updated results and add details on these additional experiments.
> > > > > >
> > > > > > **Classification**
> > > > > >
> > > > > > *MLP*: best test accuracy raised to 17.0% after random hyperparameter search with 20 seeds over the following ranges:
> > > > > > - Learning rate: [1e-5, 1e-2]
> > > > > > - Number of layers: {2,3,4,5,6}
> > > > > > - Number of channels: [16, 256]
> > > > > >
> > > > > > *Non-uniform convolution*: best test accuracy raised to 30.1% after random hyperparameter search with 10 seeds:
> > > > > > - Learning rate: [1e-5, 1e-2]
> > > > > > - Number of channels: [16, 128]
> > > > > >
> > > > > > For comparison:
> > > > > > - 2-layer DI-Net accuracy: 32.9%
> > > > > > - 2-layer CNN accuracy: 33.7%
> > > > > >
> > > > > > **Segmentation**
> > > > > >
> > > > > > *Hypernetwork*: best result on coarse segs raised to 0.059 mIoU, 14.4% pixel accuracy after random hyperparameter search with 20 seeds:
> > > > > > - Learning rate: [1e-5, 1e-2]
> > > > > > - Number of layers: {2,3,4,5,6}
> > > > > > - Number of channels: [16, 128]
> > > > > >
> > > > > > *Non-uniform convolution*: best result on coarse segs raised to 0.214 mIoU, 52.3% pixel accuracy after random hyperparameter search with 10 seeds:
> > > > > > - Learning rate: [1e-5, 1e-2]
> > > > > > - Number of channels: [8, 64]
> > > > > >
> > > > > > For comparison:
> > > > > > - 3-layer DI-Net: 0.471 mIoU, 78.5% PixAcc
> > > > > > - 3-layer CNN: 0.488 mIoU, 79.4% PixAcc

---

### Official Review · Reviewer_pEzC · 2022-10-25

**Confidence:** 3
**Correctness:** 3
**Technical Novelty And Significance:** 3
**Empirical Novelty And Significance:** 2
**Recommendation:** 6

**Clarity, Quality, Novelty And Reproducibility:**

Clarity:
The paper is hard to read and uses many mathematical concepts without clear introduction in the main text. Examples include:
  - ideal, top of page 4
  - variation and discrepancy in def 2
  - weak derivative
  - Fréchet differentiability

Quality: as far as I can tell, the mathematics are sound.

Novelty: the theoretical insights are novel, but the methodological innovation seems somewhat limited.

Reproducibility: the paper appears reproducible.

**Strength And Weaknesses:**

Strength:
- Appears a thorough and novel theoretical analysis of maps between NFs, proving desirable properties
- Gives an interesting empirical insight into how to choose discretizations.
- The method is evaluated on three different interesting problem domains.

Weaknesses:
- I'm a quite confused about the weak derivatives in eqns (3) and (6) and the assumption that the NF is in the Sobolev space. Is the existence of the weak derivatives assumed throughout? If so, that's not made very explicit. Step 2 in the sketch of the proof of Theorem 2 also only mentions function evaluation, not taking the weak derivatives. Does it matter for the universality if the DI-Net takes gradients as input? It'd be great if the authors could clarify the role of these weak gradients.
- The paper cites other methods to process NFs, such as via hypernetworks, but only compares grid discretizations to other discretizations.

**Summary Of The Paper:**

The paper proposes a general framework for building neural networks that take neural fields (NFs) - vector fields over compact spaces - as input and/or output. Processing NFs requires choosing a discretization on the compact space. The paper studies the sensitivity of maps between NFs on the choice of discretization. They define "Discretization Invariant" (DI) maps between NFs to be those for which the discretization error on a finite sample X is bounded by the variation of the input NF and the discrepancy of X.

The paper then proposes a family of DI invariant maps to be continuously defined convolution-like maps between fields on a particular discretization. The authors prove that this construction is universal and that the gradients can be estimated consistently on finite samples.

Various train/test discretizations are evaluated on classification on a NF version of imagenet, segmentation and signed distance function prediction.

**Summary Of The Review:**

The paper does a novel in-depth analysis of maps between NFs, but it is difficult to follow and lacks comparisons to baslines.

---

> ### Author Response · Authors · 2022-11-19
> **Reply to Reviewer pEzC**
>
> We thank the reviewer for their helpful suggestions. We are pleased that they found our approach to be a thorough and novel theoretical analysis with interesting empirical insight". Based on their feedback, we have made several revisions to add technical clarifications, presentation improvements, and more baselines to our experiments. We respond to their individual comments below:
>
> > "Is the existence of the weak derivatives assumed throughout?"
>
> Yes, the weak derivatives were assumed throughout. However, in our experiments we do not use layers that depend on weak derivatives, and in practice derivatives are only needed in rare cases such as total variation regularization. Therefore we remove this dependence in the main text of the revision, while showing in the appendix that our theoretical results also hold for layers that depend on weak derivatives.
>
> > “Does it matter for the universality if the DI-Net takes gradients as input?”
>
> As the reviewer notes, our universal approximation theorem does not require that layers depend on weak derivatives. Since allowing layers to depend on weak derivatives cannot reduce the expressivity of DI-Nets, they would also be universal approximators. We have edited the appendix to emphasize this point.
>
> > “The paper cites other methods to process NFs, such as via hypernetworks, but only compares grid discretizations to other discretizations”
>
> We have added comparisons to two additional baselines, including hypernetworks.
> * On classification, we added an MLP that predicts class labels directly from SIREN parameters, and added non-uniform convolutions [1] as a representative of other approaches to processing non-grid data. We find that both approaches perform significantly worse than our method and CNN baseline.
> * On segmentation, we added a hypernetwork that maps the parameters of the RGB NF to the parameters of a similar NF that represents its segmentation, as well as the non-uniform convolution in [1]. Similarly, we find that both approaches underperform.
>
> We also point out that that both of these previous approaches lack key benefits of our approach: the MLP/hypernetwork approach is not interoperable between different types of neural fields and cannot be applied to fields whose parameters extend beyond a neural network; the non-grid convolution [1] cannot preserve high-frequency information and only produces outputs along the grid unless other techniques are incorporated.
>
> [1] Jiang et al. Convolutional Neural Networks on non-uniform geometrical signals using Euclidean spectral transformation. ICLR 2019.
>
> > "The paper is hard to read and uses many mathematical concepts without clear introduction in the main text."
>
> We thank the reviewer for identifying mathematical concepts that require further explanation. We hope our significant revisions to section 3 address these concerns. These changes include:
> - *ideal*: we define the discrete map informally rather than as a quotient space
> - *variation and discrepancy*: we define them for a simple 1D case
> - *weak derivative*: we only describe the (entirely optional) dependence on weak derivatives in the appendix
> - *Gateaux differentiability* (we discovered in our revisions that this weaker condition is sufficient): we provide intuition for the Gateaux derivative and why we need to invoke it here.
>
> > “The theoretical insights are novel, but the methodological innovation seems somewhat limited”
>
> Our theoretical contributions are indeed the focus of our paper. Nevertheless we believe our paper offers interesting empirical insights, including:
> - The model often generalizes to a different discretization (in terms of sampling method and number of points) at test time (Tables 1-3, Fig 3).
> - The manner in which the model fails to generalize is task-dependent (compare Shrunk → QMC in Tables 1 and 3).
> - QMC discretizations are monotonically convergent whereas grid discretizations are only convergent in integer multiples (Fig. E.1).
> - Initializing DI-Net with weights from large pre-trained models results in unstable training behavior (Tables E.1-E.2, Fig. E.2).
>
> We hope that these empirical findings can inspire future theoretical and empirical works following up on these important practical aspects.

---

> > ### Comment · Reviewer_pEzC · 2022-11-24
> > **Thank you for the improvements in readability**
> >
> > I thank the authors for the improvement in readability of their revised version. My score remains unchanged, but I've increased my confidence.
> >
> > Small comment:
> > - The authors are now using Gateaux derivatives without a proper introduction. I suggest the authors add a clear definition.

---

> > > ### Author Response · Authors · 2022-11-24
> > > **Thank you**
> > >
> > > Thank you for your support! We will add an explicit definition of the Gateaux derivative to future versions of the paper.

---

### Author Response · Authors · 2022-11-19
**General response to reviewers**

We thank the reviewers for their thoughtful and detailed feedback which has been used to improve the submission considerably.
As a summary of the reviews, we are happy that they found our work a thorough, novel, and interesting (pEzC, 6WWK, XJni) approach to an important (6WWK) and timely (XJni) problem: how to learn maps between neural fields without being sensitive to how the field is discretized. Common concerns revolved around clarity, reproducibility, and requests for additional baselines.
We revised our paper to address these concerns, with updated text in blue. We summarize major changes and improvements based on reviewer feedback to the submission below.

### Major Changes:
- To address our usage of **unclear or overly formal language** (pEzC, 6WWK, XJni), we significantly restructured section 3 to focus on building intuition for our method and theoretical results. We make more concrete connections to design and implementation, and move technical desiderata and less important definitions to the appendix. Thanks to the reviewer’s suggestions we believe the section is much clearer and more appealing to a broader audience.
- To improve the **reproducibility** of our work (6WWK, NWbG), we attach an anonymized version of our code base in the supplementary materials, with instructions for replicating our experiments. If our paper is accepted, we will add a link to the code repository. Further, we now include more experimental details in the appendix for completeness.
- To better place our work in the literature, we added comparisons to **additional baselines** (pEzC, 6WWK, XJni). On classification, we added an MLP that predicts classification directly from SIREN parameters, and added [1] as a representative of non-uniform convolutional networks. We find that both approaches perform significantly worse than our method and the CNN baseline (MLP has 13.9% accuracy and non-uniform method has 28.9% accuracy, compared to 32.9% of our 2-layer DI-Net and 33.7% for the 2-layer CNN). On segmentation, we added a hypernetwork that maps the parameters of each SIREN image to the parameters of a new SIREN that represents its segmentation, as well as the non-uniform convolution in [1]. Similarly, we find that both approaches underperform - the hypernetwork has an mIoU of 0.038, and the non-uniform convolution has an mIoU of 0.109, compared to 0.471 for a 3-layer DI-Net. We also point out that that both of these previous approaches lack key benefits of our approach: the MLP/hypernetwork approach is not interoperable between different types of neural fields and cannot be applied to fields whose parameters extend beyond a neural network; the non-grid convolution [1] cannot preserve high-frequency information and only produces outputs along the grid unless other techniques are incorporated.

Finally, we emphasize that as this work is the first to provide a framework for learning mappings between neural fields across diverse tasks, we skew towards thorough theoretical analysis of the properties of such a model (while still presenting an empirically interesting and thorough small set of experiments as noted by Reviewers XJni and pEzC). In particular, we do not claim that the proposed model consistently empirically outperforms existing discrete approaches or is more scalable. Instead, we hope that our submission encourages future work which builds upon the theory and promising experiments presented here and scales DI-Nets to challenging real world tasks.

*References:*

[1] Jiang et al. Convolutional Neural Networks on non-uniform geometrical signals using Euclidean spectral transformation. ICLR 2019.

---

### Decision · Program_Chairs · 2023-01-20

**Decision:**

Reject

**Justification For Why Not Higher Score:**

The paper has some weakness. The numerical experiments are quite weak. It is not much convincing that the proposed method is effective in several applications. Another minor drawback is that the proposed method does not take advantage of the compact representation of neural field because it requires sampling.

**Justification For Why Not Lower Score:**

This paper's proposal seems novel. The theoretical analysis is solid. The paper is well written. Although there are some crucial weakness, the paper also gives a new insight on the literature.

**Metareview: Summary, Strengths And Weaknesses:**

This paper proposes a new modeling of neural networks called DI-Net that can process a neural field as input (and output) and has invariance against discretization. It is theoretically shown that the method works for any space discretization method as long as the discrepancy of the discretization is sufficiently small. They also proved the universal approximation ability of DI-Net. Some numerical experiments are conducted to examine the effectiveness of the proposed method.

Strength: The proposed construction of DI-Net is simple and implementable. The theoretical work (discrete sampling error and universality) seems solid and novel. The proposed framework is general so that it can be potentially useful in several applications.

Weakness:
- The numerical experiments are quite weak (as pointed by NWbG and XJni). The experiments are conducted basically on computer vision task (classification and segmentation), and a quite simple CNN baselines can be better than the proposed method. Hence, the usefulness of the proposed framework is not presented in a convincing way. It is okay that the proposed method is not always better than CNNs, and I understand the main purpose of this paper is to provide a theoretically justified framework. However, to convince the readers of the significance of the proposed framework, it is expected that numerical experiments are conducted on practically meaningful applications where the proposed method is more effective than other baseline methods (e.g., experiments of neural operators).
- (Another minor point, pointed out by XJni) The method requires space discretization and thus the merit of compact representation by the neural field can be destroyed. The authors answered that an adaptive sampling method is useful but it is not covered by the theoretical framework (the definition of invariance is given in a uniform manner over the choice of the neural field with fixed discretization).
- I (AC) also would like to point out that the convergence analysis requires that the input is in a Sobolev space but the image data is quite non-smooth so that it is not clear whether the experimental setting matches the theoretical assumptions.
- I consider (as 6WWK also pointed out) that the list of operations consisting DI-Net can be presented in a clearer way so that the concrete construction of DI-Net can be understood without ambiguity.

Although the some reviewers raised their scores during the rebuttal period, they are not strongly positive on this paper (actually, they also pointed out the drawbacks as described above). Summarizing the pros and cons, the AC considers that this paper is not well matured to be accepted in ICLR.